# Conformal Prediction under Lévy-Prokhorov Distribution Shifts: Robustness to Local and Global Perturbations

**Liviu Aolaritei**[*]
UC Berkeley
Berkeley, CA
liviu.aolaritei@berkeley.edu

**Julie Zhu**[*]
MIT
Cambridge, MA
qianyu_z@mit.edu

**Oliver Wang**[*]
MIT
Cambridge, MA
olivrw@mit.edu

**Michael I. Jordan**
UC Berkeley
Berkeley, CA
jordan@cs.berkeley.edu

**Youssef Marzouk**
MIT
Cambridge, MA
ymarz@mit.edu

## Abstract

Conformal prediction provides a powerful framework for constructing prediction intervals with finite-sample guarantees, yet its robustness under distribution shifts remains a significant challenge. This paper addresses this limitation by modeling distribution shifts using Lévy-Prokhorov (LP) ambiguity sets, which capture both local and global perturbations. We provide a self-contained overview of LP ambiguity sets and their connections to popular metrics such as Wasserstein and Total Variation. We show that the link between conformal prediction and LP ambiguity sets is a natural one: by propagating the LP ambiguity set through the scoring function, we reduce complex high-dimensional distribution shifts to manageable one-dimensional distribution shifts, enabling exact quantification of worst-case quantiles and coverage. Building on this analysis, we construct robust conformal prediction intervals that remain valid under distribution shifts, explicitly linking LP parameters to interval width and confidence levels. Experimental results on real-world datasets demonstrate the effectiveness of the proposed approach.

## 1 Introduction

Conformal prediction has emerged as a versatile framework for constructing prediction intervals with finite-sample coverage guarantees [32, 40, 2]. By leveraging the concept of nonconformity, it provides valid confidence sets for predictions, regardless of the underlying data distribution. This framework has gained significant traction in fields such as medicine [29, 38], bioinformatics [14], finance [41], and autonomous systems [27, 28], where decision-making under uncertainty is critical. However, the standard conformal prediction framework relies on the assumption of exchangeability between training and test data [4]. When this assumption is violated due to distribution shifts, the coverage guarantees of conformal prediction may break down, limiting its applicability in real-world scenarios [37].

Distribution shifts—systematic changes between the training and test distributions—are ubiquitous in practice. Examples include covariate shift in medical diagnostics, where

---

[*]: Equal contribution.

39th Conference on Neural Information Processing Systems (NeurIPS 2025).

the population characteristics evolve over time [36], or adversarial perturbations in image classification, where small, targeted changes to inputs can drastically alter predictions [31]. Addressing such shifts is essential for ensuring the reliability of predictive models, particularly in high-stakes applications.

Existing extensions of conformal prediction under distribution shifts impose restrictive structural assumptions: they assume particular types of covariate or label shift [37, 34], purely local $\ell_2$-bounded perturbations or purely global contamination [16, 11], shifts measured by a prescribed $f$-divergence [9]. While effective in certain settings, these approaches can struggle with more complex shifts that involve both local perturbations (e.g., small, pixel-level changes in images) and global perturbations (e.g., population-wide shifts in feature distributions) [6]. To bridge this gap, we propose a novel framework based on Lévy–Prokhorov (LP) ambiguity sets, a class of optimal transport-based discrepancy measures that simultaneously capture local and global perturbations.

LP ambiguity sets offer a flexible and interpretable way to model distributional uncertainty. Unlike $f$-divergences, which are limited to absolutely continuous shifts, LP metrics naturally handle broader scenarios, including discrete and transport-based perturbations [7]. For example, LP metrics can capture local shifts such as minor variations in image textures or sensor readings, as well as global shifts like changes in population demographics. This dual capability makes LP metrics particularly suited for robust prediction in dynamic and heterogeneous environments.

In this paper, we leverage the LP ambiguity set to develop a distributionally robust extension of conformal prediction. By propagating LP ambiguity sets through the scoring function, we simplify high-dimensional shifts into one-dimensional shifts in the score space, enabling exact quantification of worst-case quantiles and coverage. This approach leads to interpretable and robust prediction intervals, with explicit control over how the local and global LP parameters influence interval width and confidence levels.

Finally, we validate the proposed approach on three benchmark datasets: MNIST [25], ImageNet [13], and iWildCam [6], the latter of which captures real-world distribution shifts, demonstrating its empirical coverage guarantees and efficiency in terms of prediction set size.

## 1.1 Related Work

Under train-test distribution shifts that violate exchangeability, conformal prediction often fails to maintain valid coverage guarantees [37]. Extensions to conformal prediction under such shifts can be summarized into three main categories: sample reweighting, ambiguity sets, and sequential learning.

*Sample Reweighting.* This approach assigns weights to calibration samples based on their relevance to the test data. For instance, [37] proposed weighted conformal prediction for covariate shift, where the marginal distribution $\mathbb{P}_X$ changes while the conditional distribution $\mathbb{P}_{Y|X}$ remains fixed. Likelihood ratios are used to adjust for compositional differences, enabling valid predictions. Subsequent extensions address label shift [34], causal inference [26], and survival analysis [8, 20]. However, these methods rely on the accurate estimation of likelihood ratios, which may be challenging in practice. For spatial data, [30] proposed weighting samples based on proximity to test points. Still within the covariate shift setting, [35] and [46] leverage semiparametric theory to design more efficient conformal methods with asymptotic conditional coverage, bypassing the need for explicit sample reweighting. Compared to these approaches, our method handles distribution shifts in the *joint* distribution $\mathbb{P}$ of $(X, Y)$, without requiring likelihood ratios, and remains effective under more complex local and global perturbations.

*Ambiguity Sets.* Ambiguity sets provide a flexible framework for modeling uncertainty in the data distribution. For instance, [9] used an $f$-divergence ambiguity set around the training distribution to derive worst-case coverage guarantees and adjusted prediction sets. This work is most closely related to ours, and while their analysis inspired our approach, we rely on fundamentally different tools, particularly drawing on optimal transport techniques. A key limitation of $f$-divergences is that they are restricted to distribution shifts that are absolutely continuous with respect to the training distribution. Building on this line

of work, [1] proposed a robust conformal inference framework that explicitly separates covariate and conditional shifts: the former is handled via sample reweighting without constraints, while the latter is modeled using an $f$-divergence ball. This decomposition enables distinct handling of covariate and conditional shifts, improving efficiency compared to worst-case joint modeling. A related approach is Wasserstein-Regularized Conformal Prediction (WR-CP) [44], which heuristically minimizes an empirical upper bound on the coverage gap under joint distribution shift by combining importance weighting with Wasserstein distance regularization in score space. However, WR-CP requires kernel density estimation and repeated Wasserstein computations during training, and does not offer formal coverage guarantees under worst-case shifts. Differently, [16] proposed robust score functions based on randomized smoothing [12, 24], which ensure valid predictions under adversarial perturbations within $\ell_2$-norm balls. While adversarial methods tend to produce overly conservative uncertainty sets, recent works [45, 17, 11] have refined prediction sets by considering specific perturbation structures. Other extensions have incorporated poisoning attacks and non-continuous data types such as graphs [49]. However, these methods often assume very specific types of distribution shifts or require solving complex optimization problems. In a related spirit, both [47] and [21] study worst-case coverage under unmeasured confounding, modeled via the $\Gamma$-selection framework. While their focus is on causal inference and the distributional shifts induced by hidden confounders, their robustness guarantees parallel our LP-based approach in targeting worst-case coverage over a structured class of perturbations. In contrast, our method employs a unified discrepancy measure that captures both local and global perturbations, imposes no assumptions on the score distribution, and provides a computationally efficient way to construct prediction sets.

*Sequential Learning.* While most methods assume i.i.d. or exchangeable training data, several works have explored sequential conformal prediction. These methods include updating nonconformity scores [42], leveraging correlation structures [10], reweighting samples [43, 4], and monitoring rolling coverage [18, 19, 48, 5]. Although our method does not focus on sequential settings, extending it to this context is a promising avenue for future research.

## 1.2 Mathematical Notation

We denote by $\mathcal{P}(\mathcal{Z})$ the space of Borel probability distributions on $\mathcal{Z} := \mathcal{X} \times \mathcal{Y} \subseteq \mathbb{R}^d \times \mathbb{R}$. Given $\mathbb{P} \in \mathcal{P}(\mathcal{Z})$, we denote by $Z \sim \mathbb{P}$ the fact that the random variable $Z$ is distributed according to $\mathbb{P}$. Projection maps are denoted by $\pi$, and the indicator function of a set $\mathcal{A}$ is denoted by $\mathbb{1}\{\mathcal{A}\}$. We implicitly assume that all maps $s : \mathcal{Z} \to \mathbb{R}$ are Borel. We denote by $s_{\#}\mathbb{P}$ the pushforward of $\mathbb{P}$ via the map $s$, defined as $(s_{\#}\mathbb{P})(\mathcal{A}) := \mathbb{P}(s^{-1}(\mathcal{A}))$, for all Borel sets $\mathcal{A} \subseteq \mathcal{Z}$. Throughout the paper, $\|\cdot\|$ denotes an arbitrary norm on $\mathcal{Z}$. Given $\mathbb{P}, \mathbb{Q} \in \mathcal{P}(\mathcal{Z})$, the $\infty$-*Wasserstein distance* is defined as

$$\mathrm{W}_{\infty}(\mathbb{P}, \mathbb{Q}) := \inf \left\{ \varepsilon \geq 0 : \inf_{\gamma \in \Gamma(\mathbb{P}, \mathbb{Q})} \int_{\mathcal{Z} \times \mathcal{Z}} \mathbb{1}\{\|z_1 - z_2\| > \varepsilon\} \, \mathrm{d}\gamma(z_1, z_2) \leq 0 \right\}, \qquad (1)$$

where $\Gamma(\mathbb{P}, \mathbb{Q})$ is the set of all joint probability distributions over $\mathcal{Z} \times \mathcal{Z}$, with marginals $\mathbb{P}$ and $\mathbb{Q}$, often called transportation plans or couplings [39]. Moreover, the *Total Variation (TV) distance* is defined as

$$\mathrm{TV}(\mathbb{P}, \mathbb{Q}) := \inf_{\gamma \in \Gamma(\mathbb{P}, \mathbb{Q})} \int_{\mathcal{Z} \times \mathcal{Z}} \mathbb{1}\{\|z_1 - z_2\| > 0\} \mathrm{d}\gamma(z_1, z_2). \qquad (2)$$

At first sight, definition (2) might seem different from the more classical definition $\mathrm{TV}(\mathbb{P}, \mathbb{Q}) = \sup\{|\mathbb{P}(\mathcal{A}) - \mathbb{Q}(\mathcal{A})| : \mathcal{A} \subseteq \mathcal{Z} \text{ is a Borel set}\}$. We refer to [23, Proposition 2.24] for a proof of their equivalence. Here, we prefer definition (2), as it demonstrates that the TV distance is a special case of an optimal transport discrepancy, enabling us to leverage the extensive literature on optimal transport [39]. Finally, we denote the $\alpha$-quantile of a distribution $\mathbb{P}$ by

$$\mathrm{Quant}(\alpha; \mathbb{P}) := \inf\{s \in \mathbb{R} : \mathbb{P}(S \leq s) \geq \alpha\}.$$

## 2 Lévy–Prokhorov Distribution Shifts

We model distribution shifts as an *ambiguity set*, i.e., a ball of probability distributions

$$\mathbb{B}_{\varepsilon, \rho}(\mathbb{P}) := \left\{ \mathbb{Q} \in P(\mathcal{Z}) : \mathrm{LP}_{\varepsilon}(\mathbb{P}, \mathbb{Q}) \leq \rho \right\}, \qquad (3)$$

around the training distribution $\mathbb{P}$, constructed using the Lévy-Prokhorov (LP) pseudo-metric

$$\text{LP}_\varepsilon(\mathbb{P}, \mathbb{Q}) := \inf_{\gamma \in \Gamma(\mathbb{P}, \mathbb{Q})} \int_{\mathcal{Z} \times \mathcal{Z}} \mathbb{1}\{\|z_1 - z_2\| > \varepsilon\} \mathrm{d}\gamma(z_1, z_2). \tag{4}$$

Note that the LP pseudo-metric belongs to the general class of optimal transport discrepancies, with the particular choice of transportation cost $c(z_1, z_2) := \mathbb{1}\{\|z_1 - z_2\| > \varepsilon\}$ [7]. In this section, we provide a detailed exposition of the LP pseudo-metric and explore its expressivity in modeling significant distribution shifts. The section culminates with Proposition 2.5, where we study the propagation of $\mathbb{B}_{\varepsilon,\rho}(\mathbb{P})$ thorough the scoring function $s$, showing that the LP distribution shift can be directly considered in the one-dimensional nonconformity scores.

For more insights into the LP ambiguity set, we begin by presenting an alternative representation that decomposes it in terms of the $\infty$-Wasserstein distance and the TV distance.

**Proposition 2.1** (Decomposition of the LP ambiguity set). *The LP ambiguity set can be equivalently rewritten as*

$$\mathbb{B}_{\varepsilon,\rho}(\mathbb{P}) = \bigcup_{\widetilde{\mathbb{P}}:\, \text{W}_\infty(\mathbb{P}, \widetilde{\mathbb{P}}) \leq \varepsilon} \left\{ \mathbb{Q} \in P(\mathcal{Z}) : \text{TV}(\widetilde{\mathbb{P}}, \mathbb{Q}) \leq \rho \right\}. \tag{5}$$

All proofs of the paper are deferred to Appendix D. The decomposition in equation (5) reveals that each distribution $\mathbb{Q} \in \mathbb{B}_{\varepsilon,\rho}(\mathbb{P})$ can be constructed through a two-step procedure. First, the center distribution $\mathbb{P}$ undergoes a *local perturbation*, resulting in an intermediate distribution $\widetilde{\mathbb{P}}$ that lies within a $W_\infty$ distance of at most $\varepsilon$ from $\mathbb{P}$. This implies that each unit of mass in $\mathbb{P}$ can be arbitrarily relocated within a radius of $\varepsilon$ in $\mathcal{Z}$. Secondly, $\widetilde{\mathbb{P}}$ is subjected to a *global perturbation*, producing the final distribution $\mathbb{Q}$, which lies within a TV distance of at most $\rho$ from $\widetilde{\mathbb{P}}$. Specifically, this step entails displacing up to a fraction $\rho$ of $\widetilde{\mathbb{P}}$'s total mass to any location in the space $\mathcal{Z}$. This decomposition in (5) immediately implies that other well-known distribution shifts can be recovered as extreme cases of the LP ambiguity set $\mathbb{B}_{\varepsilon,\rho}(\mathbb{P})$.

**Corollary 2.2** (Relationship to other metrics).

   *(i)* $\mathbb{B}_{0,\rho}(\mathbb{P})$ *recovers the TV ambiguity set* $\{\mathbb{Q} \in P(\mathcal{Z}) : \text{TV}(\mathbb{P}, \mathbb{Q}) \leq \rho\}$.

   *(ii)* $\mathbb{B}_{\varepsilon,0}(\mathbb{P})$ *recovers the $\infty$-Wasserstein ambiguity set* $\{\mathbb{Q} \in P(\mathcal{Z}) : \text{W}_\infty(\mathbb{P}, \mathbb{Q}) \leq \rho\}$.

The decomposition in (5) can also be expressed in terms of random variables, which may offer a clearer understanding of the LP distribution shifts. We state this in the following proposition, which recovers [7, Theorem 2.1] using a different approach.

**Proposition 2.3** (Local and Global Perturbation). *Let $Z_1 \sim \mathbb{P}$. Then $\mathbb{Q} \in \mathbb{B}_{\varepsilon,\rho}(\mathbb{P})$ if and only if there exists a random variable $Z_2 \sim \mathbb{Q}$ of the form*

$$Z_2 \overset{d}{=} (Z_1 + N)\mathbb{1}\{B = 0\} + C\mathbb{1}\{B = 1\}, \tag{6}$$

*the random variables $N, B, C$ are as follows: $N$ represents the local perturbation, with support $\{n \in \mathcal{Z} : \|n\| \leq \varepsilon\}$, $B$ indicates whether the sample is globally perturbed or not, with $\text{Prob}(B = 1) \leq \rho$, and $C$ represents the global perturbation, following an arbitrary distribution on $\mathcal{Z}$. In particular, $Z_1, N, B$, and $C$ can all be correlated.*

Propositions 2.1 and 2.3 readily imply that the LP ambiguity set allows for distributions $\mathbb{Q}$ which are significantly different from $\mathbb{P}$, as the following remark explains.

*Remark* 2.4 (Absolute continuity). The decomposition in (5) implies that $\mathbb{B}_{\varepsilon,\rho}(\mathbb{P})$ may contain distributions that are not absolutely continuous with respect to $\mathbb{P}$. This generality is particularly valuable in settings where the test distribution assigns mass to regions unobserved during training. Such shifts are excluded under $f$-divergence ambiguity sets [9] or models that enforce bounded likelihood ratios between the test and training distributions [37].

So far, we considered the distribution shift modeled via an LP ambiguity set in the space $\mathcal{Z} = \mathcal{X} \times \mathcal{Y}$. This is in line with supervised learning, where it is more natural to consider distribution shifts in *data-space* $\mathcal{X} \times \mathcal{Y}$, as opposed to a distribution shift in the *score-space*

$s(\mathcal{X}, \mathcal{Y})$. Nonetheless, from a technical viewpoint, it is much easier to deal with an LP ambiguity set in the one-dimensional scores, due to its immediate relationship with the cumulative distribution functions and quantiles. The following proposition shows that the result of the propagation of $\mathbb{B}_{\varepsilon,\rho}(\mathbb{P})$ through $s$ is again captured by an LP ambiguity set, allowing us to effectively restrict the analysis to a distribution shift on the scores.

**Proposition 2.5** (Propagation of the LP ambiguity set). *Let the scoring function $s : \mathcal{Z} \to \mathbb{R}$ be $k$-Lipschitz over $\mathcal{Z}$, for some $k \in \mathbb{R}_+$. Then,*

$$s_{\#}\mathbb{B}_{\varepsilon,\rho}(\mathbb{P}) \subseteq \mathbb{B}_{k\varepsilon,\rho}(s_{\#}\mathbb{P}). \tag{7}$$

Proposition 2.5 requires $s$ to be Lipschitz continuous over $\mathcal{Z}$. This condition is trivially satisfied if, for instance, $s$ is continuous and $\mathcal{Z}$ is compact. In light of the inclusion (7), we focus, for the remainder of the paper, on distribution shifts over the nonconformity scores. These shifts are modeled via an LP ambiguity set $\mathbb{B}_{\varepsilon,\rho}(\mathbb{P})$, where, for simplicity, we omit the Lipschitz constant $k$ from the notation and consider $\mathbb{P}$ to be directly the distribution of $s(Z)$. Note that, in this case, all distributions inside $\mathbb{B}_{\varepsilon,\rho}(\mathbb{P})$ are supported on $\mathbb{R}$.

*Remark* 2.6 (Lipschitzness of the score function). The Lipschitz assumption in Proposition 2.5 is not required for any other theoretical results in this paper. It merely illustrates how data-space perturbations translate into score-space perturbations under a smooth scoring function. All subsequent results, including our coverage guarantees under distribution shift, are derived by modeling shift directly over the nonconformity scores. This modeling choice aligns with standard practice in conformal prediction under distribution shift (e.g., [9]), and enables our framework to accommodate arbitrarily complex, potentially non-Lipschitz score functions such as deep neural networks.

## 3 Worst-Case Quantile and Coverage

In this section we introduce and analyze the two key quantities which allow us to construct a robust prediction interval with the right coverage level for any test distribution in the LP ambiguity set. The first quantity is the *worst-case quantile*, defined below.

**Definition 3.1** (Worst-case quantile). For $\beta \in [0, 1]$, the worst-case $\beta$-quantile in $\mathbb{B}_{\varepsilon,\rho}(\mathbb{P})$ is defined as

$$\mathrm{Quant}_{\varepsilon,\rho}^{\mathrm{WC}}(\beta; \mathbb{P}) := \sup_{\mathbb{Q} \in \mathbb{B}_{\varepsilon,\rho}(\mathbb{P})} \mathrm{Quant}(\beta; \mathbb{Q}). \tag{8}$$

Equation (8) defines the worst-case quantile through a distributionally robust optimization problem, which quantifies the largest $\beta$-quantile for all the test distributions in $\mathbb{B}_{\varepsilon,\rho}(\mathbb{P})$. In other words, $\mathrm{Quant}_{\varepsilon,\rho}^{\mathrm{WC}}(\beta; \mathbb{P})$ represents the worst-case impact of the distribution shift on the value of the $\beta$-quantile. This, in turn, affects the size of the confidence interval, as we will show in Section 4. The second quantity is the *worst-case coverage*, defined next.

**Definition 3.2** (Worst-case coverage). Let $F_{\mathbb{Q}} : \mathbb{R} \to [0, 1]$ be the cumulative distribution function of $\mathbb{Q}$.or $q \in \mathbb{R}$. Then, the worst-case coverage in $\mathbb{B}_{\varepsilon,\rho}(\mathbb{P})$ at $q$ is defined as

$$\mathrm{Cov}_{\varepsilon,\rho}^{\mathrm{WC}}(q; \mathbb{P}) := \inf_{\mathbb{Q} \in \mathbb{B}_{\varepsilon,\rho}(\mathbb{P})} F_{\mathbb{Q}}(q). \tag{9}$$

Equation (3.2) defines the worst-case coverage as the lowest value among the cumulative distribution functions in the LP ambiguity set evaluated at $q \in \mathbb{R}$. For example, if $q = \mathrm{Quant}(1 - \alpha; \mathbb{P})$, $\mathrm{Cov}_{\varepsilon,\rho}^{\mathrm{WC}}(q; \mathbb{P})$ represents the worst-case impact of the distribution shift on the true confidence level when the confidence level for $\mathbb{P}$ is $1 - \alpha$. In the remainder of this section, we will show that both $\mathrm{Quant}_{\varepsilon,\rho}^{\mathrm{WC}}(\beta; \mathbb{P})$ and $\mathrm{Cov}_{\varepsilon,\rho}^{\mathrm{WC}}(q; \mathbb{P})$ can be quantified in closed-form, as a function of the training distribution $\mathbb{P}$ and the two robustness parameters $\varepsilon, \rho$. Before doing so, we note that a high value of $\rho$, i.e., the global perturbation parameter, renders the worst-case quantile trivial. We show this in the following remark.

*Remark* 3.3 (Case $\rho \geq 1 - \beta$). If $\rho \geq 1 - \beta$, then $\mathrm{Quant}_{\varepsilon,\rho}^{\mathrm{WC}}(\beta; \mathbb{P}) = \mathrm{Quant}(1; \mathbb{P})$. Intuitively, the LP ambiguity set $\mathbb{B}_{\varepsilon,\rho}(\mathbb{P})$ allows to displace $\rho$ mass from the distribution $\mathbb{P}$ and move it arbitrarily in $\mathbb{R}$. Since $\rho \geq 1 - \beta$, this implies that we can construct a sequence of distributions $\mathbb{Q}_n \in \mathbb{B}_{\varepsilon,\rho}(\mathbb{P})$ for which $\mathrm{Quant}(\beta; \mathbb{Q}_n) \to \infty$. To see this, let $\mathbb{P} = \mathcal{U}([0, 1])$, and let $\mathbb{Q}_n := \mathcal{U}([0, 1 - \rho]) + \rho \delta_n$. Then, clearly $\mathrm{LP}_{\varepsilon}(\mathbb{P}, \mathbb{Q}_n) = \rho$, and $\mathrm{Quant}(\beta; \mathbb{Q}_n) \geq n$.

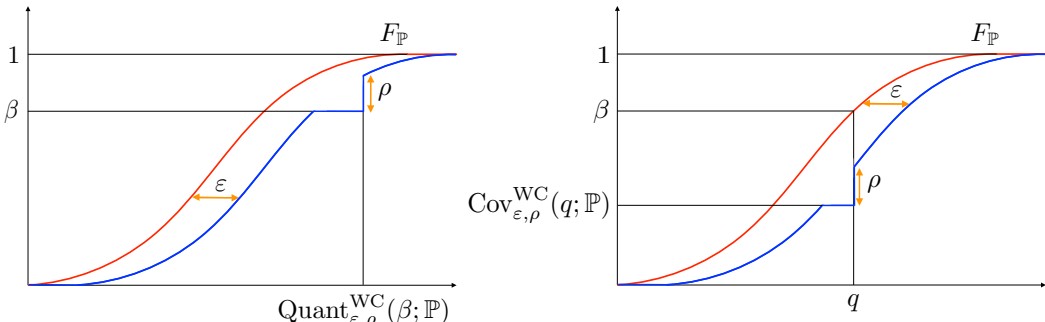

Figure 1: (Left) Worst-case quantile; (Right) Worst-case coverage.

Following Remark 3.3, we restrict our attention to the case $\rho < 1 - \beta$ in the quantity $\text{Quant}_{\varepsilon,\rho}^{\text{WC}}(\beta;\mathbb{P})$. We are now prepared to present the first result of this section.

**Proposition 3.4** (Worst-case quantile in the LP ambiguity set). *The following holds*

$$\text{Quant}_{\varepsilon,\rho}^{\text{WC}}(\beta;\mathbb{P}) = \text{Quant}(\beta + \rho;\mathbb{P}) + \varepsilon. \tag{10}$$

In words, the worst-case quantile in the LP ambiguity set $\mathbb{B}_{\varepsilon,\rho}(\mathbb{P})$ corresponds to a quantile of $\mathbb{P}$ that is shifted by the local parameter $\varepsilon$ and adjusted by the global parameter $\rho$. We will now present the second result of this section.

**Proposition 3.5** (Worst-case coverage in the LP ambiguity set). *The following holds*

$$\text{Cov}_{\varepsilon,\rho}^{\text{WC}}(q;\mathbb{P}) = F_{\mathbb{P}}(q - \varepsilon) - \rho. \tag{11}$$

The worst-case coverage in the LP ambiguity set $\mathbb{B}_{\varepsilon,\rho}(\mathbb{P})$ corresponds to the coverage of $\mathbb{P}$ shifted by the local parameter $\varepsilon$ and adjusted by the global parameter $\rho$. The proofs of Propositions 3.4 and 3.5 are constructive, in the sense that we propose two sequences of distributions which attain, in the limit, the two quantities $\text{Quant}_{\varepsilon,\rho}^{\text{WC}}(\beta;\mathbb{P})$ and $\text{Cov}_{\varepsilon,\rho}^{\text{WC}}(q;\mathbb{P})$, respectively. The intuition for both constructions stems from Proposition 2.1, which allows us to construct every distribution in $\mathbb{B}_{\varepsilon,\rho}(\mathbb{P})$ using a two-step procedure that decouples the local and global perturbations. This intuition is illustrated in Figure 1.

## 4 Distributionally Robust Conformal Prediction

In this section, we demonstrate how the worst-case quantile and coverage introduced earlier enable the construction of a confidence interval and its worst-case coverage for all distributions in the LP ambiguity set. We start by defining the prediction set

$$C_{\varepsilon,\rho}^{1-\alpha}(x;\mathbb{P}) := \left\{ y \in \mathcal{Y} : s(x,y) \leq \text{Quant}_{\varepsilon,\rho}^{\text{WC}}(1-\alpha;\mathbb{P}) \right\}, \tag{12}$$

where, as noted in Proposition 3.4, $\text{Quant}_{\varepsilon,\rho}^{\text{WC}}(1-\alpha;\mathbb{P}) = \text{Quant}(1-\alpha+\rho;\mathbb{P}) + \varepsilon$. Observe that $C_{\varepsilon,\rho}(x;\mathbb{P})$ depends on the training distribution $\mathbb{P}$, which is unknown. Instead, we assume access to $n$ exchangeable data points $\{s(X_i,Y_i)\}_{i=1}^{n} \sim \mathbb{P}$. Based on this, we define the empirical distribution $\widehat{\mathbb{P}}_n := \frac{1}{n}\sum_{i=1}^{n} \delta_{s(X_i,Y_i)}$, and consider the empirical confidence set $C_{\varepsilon,\rho}^{1-\alpha}(x;\widehat{\mathbb{P}}_n)$. We now state the main result of this paper.

**Theorem 4.1** (Conformal Prediction under LP distribution shifts). *Let $s(X_{n+1},Y_{n+1}) \sim \mathbb{P}_{\text{test}}$ be independent of $\{s(X_i,Y_i)\}_{i=1}^{n} \sim \mathbb{P}$. Moreover, let $\text{LP}_\varepsilon(\mathbb{P},\mathbb{P}_{\text{test}}) \leq \rho$. Then,*

$$\text{Prob}\left\{ Y_{n+1} \in C_{\varepsilon,\rho}^{1-\alpha}\left(X_{n+1};\widehat{\mathbb{P}}_n\right) \right\} \geq \frac{\lceil n(1-\alpha+\rho)\rceil}{n+1} - \rho. \tag{13}$$

A few remarks are in order. First, the local parameter $\varepsilon$ affects only the size of the confidence interval, but not its coverage guarantee. This is expected, given the construction of the two

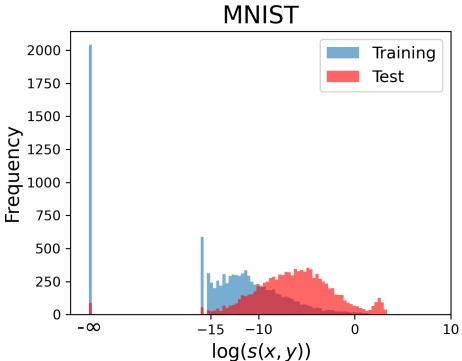
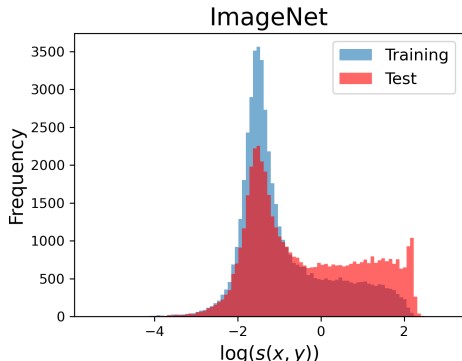

Figure 2: **Score distribution shift**. Plots for MNIST and ImageNet under ($p = 0.05, u = 1.0$) perturbation. The score distribution obtained from the unperturbed data (red), and from the perturbed data (blue) are plotted in log scale.

sequences of distributions that achieve the worst-case quantile and coverage in Propositions 3.4 and 3.5, respectively (also illustrated in Figure 1). In contrast, the global shift parameter $\rho$ influences both the coverage and the size of the prediction set: it shifts the quantile level from $1 - \alpha$ to $1 - \alpha + \rho$, and appears subtractively in the coverage bound. This change in quantile level often has a more pronounced effect on the size of the prediction set than the additive $\varepsilon$ term, particularly when the score distribution is light-tailed. Meanwhile, the reduction in coverage due to $\rho$ decreases with the calibration size $n$, and becomes negligible in the large-sample regime, scaling as $\mathcal{O}(1/n)$. Finally, as expected, the distribution shift reduces the coverage below the desired $1 - \alpha$ level. The following corollary provides an adjusted coverage for the worst-case quantile, ensuring a $1 - \alpha$ confidence level in (13).

**Corollary 4.2** ($1 - \alpha$ coverage). *Let $\beta = \alpha + (\alpha - \rho - 2)/n$. Under the same conditions as in Theorem 4.1, we have*

$$\text{Prob}\left\{Y_{n+1} \in C_{\varepsilon,\rho}^{1-\beta}\left(X_{n+1}; \widehat{\mathbb{P}}_n\right)\right\} \geq 1 - \alpha. \tag{14}$$

Recall from Corollary 2.2 that the LP pseudo-metric recovers the TV and $\infty$-Wasserstein distances if $\varepsilon = 0$ and $\rho = 0$, respectively. As a consequence, the guarantee in Corollary 4.2 can be immediately specialized to these additional types of distribution shifts.

## 5 Experiments

We conduct experiments on three classification problems: MNIST [25], ImageNet [13], and iWildCam [6]. We also compare our algorithm against five other methods: standard split conformal prediction (SC), $\chi^2$-divergence robust conformal prediction [9], conformal prediction under covariate shift (Weight) [37], randomly smoothed conformal prediction (RSCP) [16], and fine-grained conformal prediction (FG-CP) [1]. Each method defines its own prediction set; for our method, this is the robust set $C_{\varepsilon,\rho}^{1-\beta}(x; \widehat{\mathbb{P}}_n)$ from Corollary 4.2. While additional methods exist in the literature, they typically constitute minor variations or special cases of the five representative baselines we benchmark against.

We evaluate methods in terms of *validity* and *efficiency*. Validity is computed as the average empirical coverage across $M$ independent calibration-test splits $\frac{1}{M} \sum_{j=1}^{M} [\frac{1}{K} \sum_{i=1}^{K} \mathbb{1}\{y_i^{(j)} \in C(x_i^{(j)}; \widehat{\mathbb{P}}_n^{(j)})\}]$, where $\widehat{\mathbb{P}}_n^{(j)}$ denotes the empirical distribution of the $j$-th calibration set, and $\{(x_i^{(j)}, y_i^{(j)})\}_{i=1}^{K}$ denotes the corresponding test set. Efficiency is evaluated as the average prediction set size across the same $M$ splits and $K$ test samples. For all experiments, we set the miscoverage level to $\alpha = 0.1$ and use the negative log-likelihood (NLL) score, $s(x, y) = -\log p(y \mid x)$, as the nonconformity measure. For ImageNet, we use a pre-trained ResNet-152 model; for MNIST, we train a small ResNet architecture from scratch; and for iWildCam, we adopt the pre-trained ResNet-50 model provided by [6].

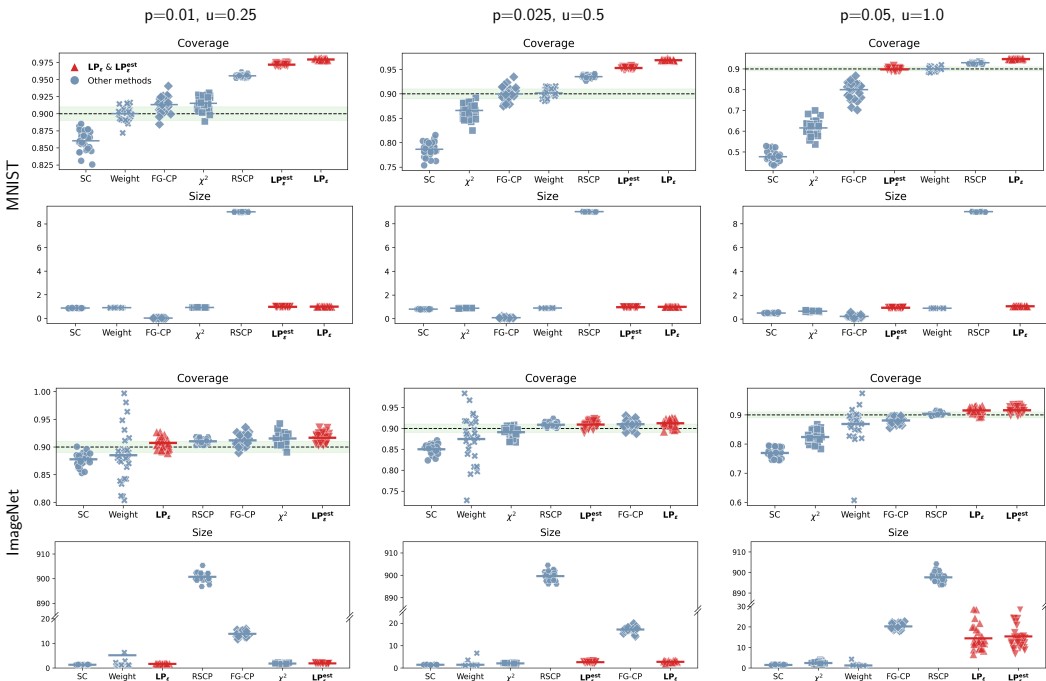

Figure 3: **MNIST and ImageNet**. Coverage (validity) and size (efficiency). In the coverage plots, the long dashed line indicates the target $1 - \alpha$ level. Scattered points show empirical coverage and prediction set size for each calibration–test split, while short horizontal lines denote averages across $M = 30$ splits. The proposed methods are highlighted in bold/red.

## 5.1 Data-Space Distribution Shift: MNIST and ImageNet

Following the split conformal procedure, we partition the hold-out validation set into a calibration set of $n = 1000$ samples and a test set of $K = 5000$ samples drawn uniformly from the remaining data. We simulate local perturbations by adding i.i.d. noise from $\mathcal{U}([-u, u])$ to every channel of each test image. Global perturbations are introduced by randomly corrupting a fraction $p$ of test labels, replacing each with a neighboring class label. This setup captures realistic scenarios in which test-time inputs are noisy and some labels may be incorrect due to annotation errors [15, 49]. Figure 2 illustrates the resulting shift in the score distribution under the perturbation setting ($p = 0.05, u = 1.0$).

Calibration NLL scores are computed on unperturbed calibration data points to determine empirical quantiles. Constructing prediction sets is then straightforward for standard conformal prediction. For the robust algorithms, our method naturally accounts for both global and local perturbations through the parameters $\rho$ and $\varepsilon$, respectively. Following Proposition 2.5, we set $\rho = p$ to reflect the global label corruption level. While the same proposition suggests setting $\varepsilon = ku$, where $k$ is the Lipschitz constant of the score function, estimating $k$ from data often leads to overly conservative values, as a global Lipschitz constant may not reflect the local behavior of the score function where the data are concentrated. In practice, we find that a fixed value $k = 2$ suffices to ensure valid coverage across the full range of data-space shifts $u$; we refer to this method as $\mathrm{LP}_\varepsilon$. In parallel, we evaluate a data-driven variant, called $\mathrm{LP}_\varepsilon^{\mathrm{est}}$, which estimates both $\varepsilon$ and $\rho$ directly from samples using the algorithm described in Appendix B. This version achieves similar robustness while adapting more flexibly to the underlying shift. For the $\chi^2$, FG-CP, RSCP, and Weight conformal prediction methods, we follow the original experimental setups described in their respective references; implementation details are provided in Appendix C.

Figure 3 reports the empirical coverage and prediction set size (averaged over 30 calibration–test splits) for the seven methods under three levels of noise corruption: $(p, u) \in \{(0.01, 0.25), (0.025, 0.5), (0.05, 1.0)\}$. As expected, standard conformal prediction (SC) fails

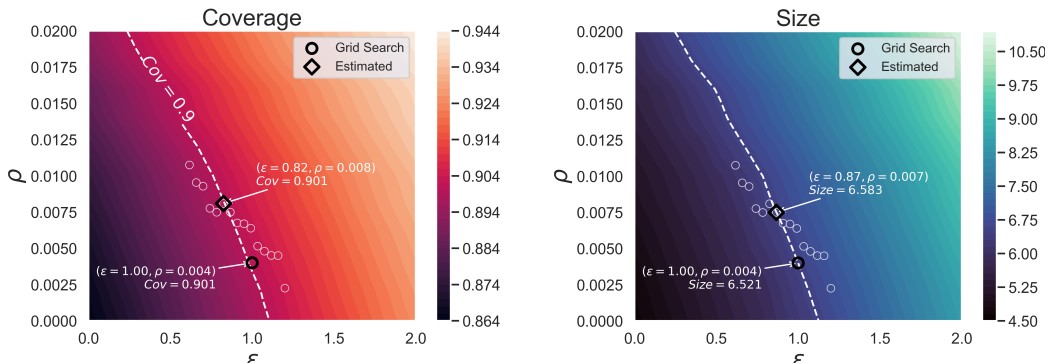

Figure 4: **iWildCam.** Coverage (left) and prediction set size (right) across $(\varepsilon, \rho)$ values. The white dashed line marks the set of $(\varepsilon, \rho)$ pairs achieving exactly 90% empirical coverage, and the smallest prediction set along this frontier is shown with a black circle. White circles correspond to points estimated by the algorithm in Appendix B, and the best-performing pair among them (yielding the smallest prediction set) is marked by a black diamond.

to maintain coverage as the corruption level increases. In contrast, both variants of our method—$\mathrm{LP}_\varepsilon$ and $\mathrm{LP}_\varepsilon^{\mathrm{est}}$—consistently maintain valid coverage across all settings. They also achieve comparable prediction set sizes, demonstrating the effectiveness of data-driven parameter estimation. Among the remaining baselines, only RSCP maintains valid coverage under all shift levels, but it does so at the cost of extremely large prediction sets, particularly for ImageNet. The other three baselines, i.e., $\chi^2$, FG-CP, and Weight, exhibit coverage degradation as the shift intensity increases. This is expected: these methods assume absolute continuity between the training and test distributions, a condition violated in our experimental setup (see Figure 2). In particular, when test-time perturbations cause the support of the test distribution to lie partially outside that of the training distribution, methods relying on importance weighting or $f$-divergence balls struggle to provide valid guarantees. In contrast, our LP-based approach requires no absolute continuity and remains robust to both global label corruption and local input noise.

This numerical illustration also highlights an important *modeling* point. The LP-based approach is specifically designed to capture local and global perturbations of the data distribution, as introduced in this experiment. It provides a principled framework for handling such shifts, complete with closed-form expressions for both the worst-case quantile and coverage. As a result, the strong empirical performance observed here is not coincidental: our method is theoretically tailored to this class of distribution shifts, and no other method can offer stronger worst-case guarantees within the same ambiguity set.

## 5.2 Real-world Distribution Shift: iWildCam

We now evaluate our algorithm's ability to handle real-world distribution shifts using the iWildCam dataset [6], a multi-class classification task characterized by naturally occurring train-test discrepancies. These arise from changes in camera trap placement and timing, which induce variability in illumination, color, viewpoint, background, vegetation, and species frequency. As described in [22], the dataset includes a training set, an out-of-distribution test set, and an in-distribution validation/test set consisting of images captured from the same camera locations as the training data but on different dates. We use the in-distribution test set for calibration and the out-of-distribution test set for evaluation.

Figure 5.2 illustrates how coverage and prediction set size vary over a grid of $(\varepsilon, \rho)$ values in the LP ambiguity set. The left panel shows that all pairs lying to the right of the black dotted contour (the 90% coverage curve) yield valid coverage under the real distribution shift. This demonstrates that LP ambiguity sets capture the relevant perturbations affecting iWildCam, without assuming prior knowledge of the shift type or structure. The right panel shows the corresponding prediction set sizes. Notably, moving further right from the 90%

contour leads to increasingly conservative sets. White circles in both panels denote $(\varepsilon, \rho)$ pairs estimated by the data-driven procedure described in Appendix B. The best among these—marked with a black diamond—achieves nearly identical coverage and prediction set size as the optimal point found by an exhaustive grid search (marked by a black circle). This proximity confirms that the proposed estimation algorithm reliably recovers high-quality ambiguity set parameters with limited test data.

Taken together, these results support two key takeaways: (1) LP ambiguity sets flexibly model real distribution shifts, delivering valid coverage across a broad region of the parameter space, and (2) the estimated $(\varepsilon, \rho)$ pair performs comparably to the best grid-tuned pair, both in coverage and efficiency.

## Acknowledgement

Liviu Aolaritei acknowledges support from the Swiss National Science Foundation through the Postdoc.Mobility Fellowship (grant agreement P500PT_222215). Michael Jordan was funded by the Chair "Markets and Learning," supported by Air Liquide, BNP PARIBAS ASSET MANAGEMENT Europe, EDF, Orange and SNCF, sponsors of the Inria Foundation. Youssef Marzouk and Julie Zhu acknowledge support from the US Department of Energy (DOE), Office of Advanced Scientific Computing Research, under grant DE-SC0023188. Youssef Marzouk and Zheyu Oliver Wang acknowledge support from the ExxonMobil Technology and Engineering Company.

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

# A  Preliminaries in Conformal Prediction

In what follows, we provide a brief introduction to *split conformal prediction*. Consider a predictive model $f : \mathcal{X} \to \mathcal{Y}$ and a calibration dataset $\mathcal{D} = \{(X_i, Y_i)\}_{i=1}^n \subseteq \mathcal{X} \times \mathcal{Y}$, where the points in $\mathcal{D}$, along with any test sample $(X_{n+1}, Y_{n+1}) \in \mathcal{X} \times \mathcal{Y}$, are assumed to be exchangeable and distributed according to $\mathbb{P}$. Without additional assumptions on the predictive model or the data-generating process, conformal prediction constructs a prediction set $C^{1-\alpha}(X_{n+1})$ that satisfies the finite-sample coverage guarantee:

$$\text{Prob}\left\{Y_{n+1} \in C^{1-\alpha}(X_{n+1})\right\} \geq 1 - \alpha, \tag{15}$$

where the probability is taken over both the calibration dataset $\mathcal{D}$ and the test point $(X_{n+1}, Y_{n+1})$.

To achieve this, conformal prediction relies on a scoring function $s : \mathcal{X} \times \mathcal{Y} \to \mathbb{R}$, which quantifies the nonconformity of a label $y \in \mathcal{Y}$ for a given input $x \in \mathcal{X}$. The predictive model $f$ is typically used to define the scoring function $s$, where $f(x)$ represents the model's prediction. In regression, $f(x)$ might return a point estimate of $y$, with $s(x, y)$ defined as the absolute error $|f(x) - y|$. In classification, $f(x)$ might output class probabilities, and $s(x, y)$ could be the negative log-probability of the true label $y$. For each calibration point $(X_i, Y_i) \in \mathcal{D}$, the nonconformity score $s(X_i, Y_i)$ is computed. The scores are then used to estimate the empirical $(1 - \alpha)$-quantile, with a *finite-sample correction*:

$$\widehat{q}_\alpha := \text{Quant}\left(\frac{\lceil(1-\alpha)(n+1)\rceil}{n}; s_\#\widehat{\mathbb{P}}_n\right),$$

where $s_\#\widehat{\mathbb{P}}_n$ is the empirical distribution of the calibration scores $\{s(X_i, Y_i)\}_{i=1}^n$. Finally, the prediction set for a new label $Y_{n+1}$ is defined as

$$C^{1-\alpha}(X_{n+1}) = \left\{y \in \mathcal{Y} : s(X_{n+1}, y) \leq \widehat{q}_\alpha\right\}.$$

By construction, the prediction set $C^{1-\alpha}(X_{n+1})$ satisfies the coverage guarantee in (15), provided the data is exchangeable. With the conformal prediction framework in place, we now shift our focus to the challenge of distribution shifts. Specifically, we consider scenarios where the test data $(X_{n+1}, Y_{n+1})$ is drawn from a distribution that differs from the distribution $\mathbb{P}$, with this shift captured by the Lévy–Prokhorov ambiguity set around $\mathbb{P}$. Such shifts introduce additional complexities in ensuring the robustness of the prediction intervals.

# B  Estimation of the LP ambiguity set parameters $\varepsilon$ and $\rho$

While our theoretical results apply to any pair $(\varepsilon, \rho)$ defining an LP ambiguity set, selecting these parameters in practice is critical to balancing robustness and informativeness. This is particularly important when only a limited number of calibration and test samples are available. To address this, we propose a systematic estimation procedure for $(\varepsilon, \rho)$ based on empirical data. The key idea is to identify the pair that yields the tightest worst-case conformal prediction set while preserving the desired coverage under distribution shift.

The procedure works as follows. Given two independent batches of calibration scores from the training distribution $\mathbb{P}$ and a batch of test scores from the shifted distribution $\mathbb{Q}$, we evaluate a grid of candidate $\varepsilon$ values. For each candidate $\varepsilon_i$, we estimate the corresponding $\rho_i$ by computing the LP distance between one batch of calibration scores and the test scores using one-dimensional optimal transport with cost function $\mathbb{1}\{|x - y| \geq \varepsilon_i\}$. This transport problem can be efficiently solved either via the Sinkhorn algorithm or using the standard linear programming formulation, both of which are efficient in one dimension due to the sorted structure of empirical distributions [33]. The resulting pair $(\varepsilon_i, \rho_i)$ defines a valid ambiguity set, and we compute its associated worst-case quantile using the second calibration batch. Specifically, we apply Corollary 4.2, setting $\beta_i = \alpha + (\alpha - \rho_i - 2)/n$, so that the prediction set $C_{\varepsilon_i, \rho_i}^{1-\beta_i}$ enjoys a worst-case coverage guarantee of at least $1 - \alpha$. We then select the pair that yields the smallest such quantile. To preserve statistical validity, the calibration scores used to estimate $(\varepsilon, \rho)$ must be disjoint from those used to compute the conformal quantile. This ensures that the ambiguity set is selected independently of the scores used for

---
**Algorithm 1** Estimation of $\varepsilon$ and $\rho$
---
**Input:** Independent empirical calibration score distributions $\widehat{\mathbb{P}}_n^{(1)}, \widehat{\mathbb{P}}_n^{(2)}$ and empirical test score distribution $\widehat{\mathbb{Q}}_m$; a grid of $\{\varepsilon_i\}_{i=1}^k$ values, with $k \in \mathbb{N}$; and target coverage $1 - \alpha$.
**Output:** Pair $(\varepsilon_i, \rho_i)$ yielding the tightest prediction set with valid coverage.
  1: **for** $i = 1, \ldots, k$ **do**
  2:     Compute one-dimensional LP distance $\rho_i$ between $\widehat{\mathbb{P}}_n^{(1)}$ and $\widehat{\mathbb{Q}}_m$
  3:     Set $\beta_i := \alpha + (\alpha - \rho_i - 2)/n$
  4:     Compute worst-case quantile:
  5:         $q_i := \text{Quant}_{\varepsilon_i, \rho_i}^{\text{WC}}\left(1 - \beta_i; \widehat{\mathbb{P}}_n^{(2)}\right) = \text{Quant}\left(1 - \beta_i + \rho_i; \widehat{\mathbb{P}}_n^{(2)}\right) + \varepsilon_i$
  6: **end for**
  7: **return** $(\varepsilon_i, \rho_i)$ with minimal $q_i$        ▷ Smaller $q_i$ leads to smaller robust prediction sets
---

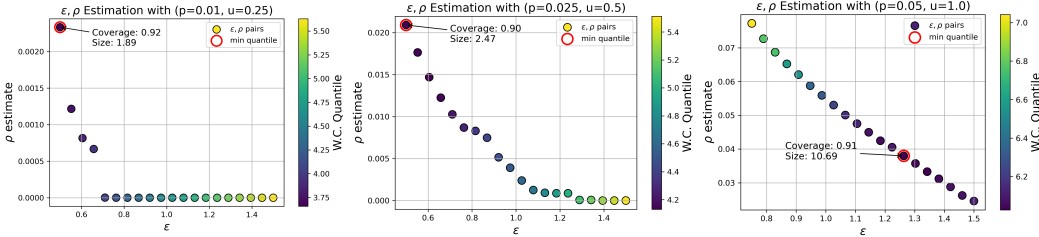

Figure 5: **ImageNet** $(\varepsilon, \rho)$ **estimation**. Each point in the 20-point grid corresponds to a candidate $(\varepsilon, \rho)$ pair, where $\varepsilon \in (0.5, 1.5)$ and $\rho$ is estimated using one-dimensional optimal transport between the empirical calibration and test score distributions, each constructed from 1000 samples. The color scale represents the empirical worst-case quantile associated with each pair, computed on a held-out calibration batch. The optimal $(\varepsilon, \rho)$ pair, yielding the smallest quantile, is highlighted in red, with the corresponding empirical coverage and prediction set size annotated. The true corruption parameters $(p, u)$ used to generate the test distribution are also indicated for reference.

calibration, avoiding overfitting and maintaining the coverage guarantee. We present this procedure in Algorithm 1.

Empirical results on ImageNet and MNIST validate the effectiveness of this approach. Figures 5 and 6 display the estimated $(\varepsilon, \rho)$ values over a 20-point grid, visualizing the resulting worst-case quantiles through color shading. The selected pair (highlighted in red) yields the smallest worst-case quantile and corresponds to the tightest robust prediction set. Across both datasets, we observe that the data-driven procedure reliably identifies ambiguity set parameters that balance coverage and informativeness, leading to prediction sets that respect the desired $1 - \alpha$ coverage level.

*Remark* B.1 (Sensitivity to $\varepsilon$ and $\rho$). It is natural to ask how sensitive the method is to misspecification of the shift parameters $(\varepsilon, \rho)$. While both influence the prediction set, their effects are asymmetric. The parameter $\varepsilon$ appears additively in the worst-case quantile and controls the width of the prediction set without affecting coverage. In contrast, $\rho$ shifts the quantile level and also appears subtractively in the coverage bound from Theorem 4.1. As a result, even small underestimations of $\rho$ can significantly impact coverage, whereas modest underestimations of $\varepsilon$ tend to reduce the prediction set size only slightly. In both Figures 5 and 6, we observe a trade-off: smaller $\varepsilon$ values are typically associated with larger $\rho$ estimates, and vice versa. Selecting the pair that minimizes the worst-case quantile provides a principled way to balance robustness and efficiency without being overly conservative.

*Remark* B.2 (Use of test samples for shift estimation). The estimation procedure outlined in this section requires access to test samples in order to estimate the distribution shift. While this may initially seem restrictive, we emphasize that only a relatively small number of test samples is needed to ensure stable estimates of $(\varepsilon, \rho)$ in practice. In our experiments, as few as 500–1000 calibration and test scores are sufficient to obtain consistent estimates across multiple runs. Nonetheless, one might ask: if test samples are available, why not apply

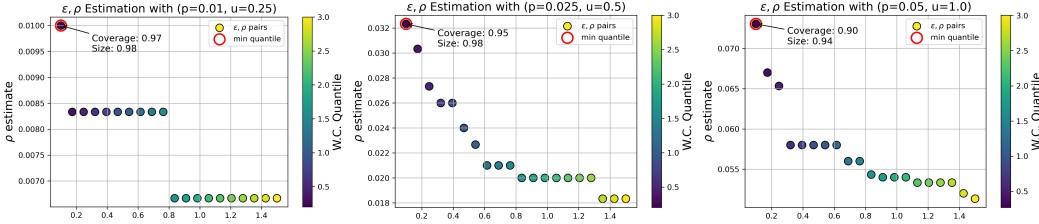

Figure 6: **MNIST** $(\varepsilon, \rho)$ **estimation**. Each point in the 20-point grid corresponds to a candidate $(\varepsilon, \rho)$ pair, where $\varepsilon \in (0.1, 1.5)$ and $\rho$ is estimated using one-dimensional optimal transport between the empirical calibration and test score distributions, each constructed from 1000 samples. The color scale represents the empirical worst-case quantile associated with each pair, computed on a held-out calibration batch. The optimal $(\varepsilon, \rho)$ pair, yielding the smallest quantile, is highlighted in red, with the corresponding empirical coverage and prediction set size annotated. The true corruption parameters $(p, u)$ used to generate the test distribution are also indicated for reference.

conformal prediction directly to them instead of using a distributionally robust approach? In many applications, this is indeed preferable, as standard conformal methods yield valid coverage guarantees under the exchangeability assumption. The purpose of this section, however, is not to recommend distributionally robust conformal prediction over standard conformal prediction in the presence of test data. Rather, it is to demonstrate LP-based ambiguity sets as a principled model for capturing both local and global distribution shifts. Estimating these parameters from data allows us to instantiate the LP ambiguity set in a concrete, data-driven way. Nonetheless, we acknowledge as a *limitation* of our current approach that it requires access to test samples for estimating the shift. Developing estimators that rely solely on calibration data—such as the variability-based method proposed by [9]—is an important direction for future work.

## C   Experimental Setup

All experiments were conducted on a single Nvidia A100 GPU with 40GB of RAM. We strictly follow the official GitHub implementations provided by the authors of the referenced methods, except for weighted conformal prediction [37], for which we implemented a neural network–compatible version based strictly on the algorithm described in [37].

For a given level $\alpha$ and $n$ calibration data points, the prediction sets for each algorithm are constructed from the following quantiles:

1. *Standard Conformal Prediction:*
$$\text{Quant}\left(\lceil (n+1)(1-\alpha)\rceil/n; \widehat{\mathbb{P}}_n\right)$$

2. *Our method—LP Robust Conformal Prediction (following Corollary 4.2):*
$$\text{Quant}\left(1-\beta+\rho; \widehat{\mathbb{P}}_n\right)+\varepsilon, \qquad \beta = \alpha + (\alpha - \rho - 2)/n$$

3. $\chi^2$ *Robust Conformal Prediction [9]:*
$$\text{Quant}\left(g_{f,\rho}^{-1}(1-\alpha_n); \widehat{\mathbb{P}}_n\right), \ \alpha_n = g_{f,\rho}\left((1+1/n)\, g_{f,\rho}^{-1}(1-\alpha)\right),$$

   where $\rho$ is the radius of the ambiguity set, $f(x) = (x-1)^2$, and $g_{f,\rho}$ and $g_{f,\rho}^{-1}$ are defined as:
$$g_{f,\rho}(\beta) := \inf\left\{z \in [0,1] : \beta f\left(\frac{z}{\beta}\right) + (1-\beta)f\left(\frac{1-z}{1-\beta}\right) \le \rho\right\}, \ \beta \in [0,1],$$
$$g_{f,\rho}^{-1}(\tau) = \sup\left\{\beta \in [0,1] : g_{f,\rho}(\beta) \le \tau\right\}, \ \tau \in [0,1].$$

   The radius $\rho$ is estimated using the slab estimation procedure described in [9].

4. *Conformal Prediction under Covariate Shift [37]:*

$$\text{Quant}\left(1 - \alpha; \widehat{\mathbb{P}}_{n+1}^{\omega}\right), \qquad \widehat{\mathbb{P}}_{n+1}^{\omega} := \sum_{i=1}^{n} p_i^{\omega}(x)\delta_{s(X_i, Y_i)} + p_{n+1}^{\omega}(x)\delta_{\infty}$$

where the weights are defined by

$$p_i^{\omega}(x) = \frac{\omega(X_i)}{\sum_{j=1}^{n} \omega(X_j) + \omega(x)}, \ i = 1, \dots, n, \qquad p_{n+1}^{\omega}(x) = \frac{\omega(x)}{\sum_{j=1}^{n} \omega(X_j) + \omega(x)}.$$

Here, $\omega(X) = \mathrm{d}\mathbb{P}_{\text{test}}(X)/\mathrm{d}\mathbb{P}_{\text{calib}}(X)$ denotes the density ratio between the test and calibration distributions, estimated via a separately trained classifier.

5. *Randomly Smoothed Conformal Prediction [16]:*

$$\text{Quant}\left((1 - \alpha)(2 + n)/(1 + n); \widetilde{\mathbb{P}}_n\right) + \delta/\sigma, \qquad \widetilde{\mathbb{P}}_n := \frac{1}{n}\sum_{i=1}^{n}\delta_{\tilde{s}(X_i, Y_i)}.$$

Here, $\tilde{s}$ denotes the smoothed nonconformity score based on an existing score function $s$ [16], under which adversarial noise with $\|\epsilon\|_2 \leq \delta$ is propagated with distortion bounded by $\delta/\sigma$.

6. *Fine-grained Conformal Prediction [1]:*

$$\text{Quant}\left(g_{f,\rho}^{-1}(1 - \alpha), \widehat{\mathbb{P}}_{n+1}^{\omega}\right)$$

where $g_{f,\rho}^{-1}$ and $\widehat{\mathbb{P}}_{n+1}^{\omega}$ are defined above. For the $f$-divergence method, we estimate the robustness parameter $\rho$ using the slab estimation procedure described in [9].

## D  Proofs

### D.1  Proofs of Section 2

#### D.1.1  Proof of Proposition 2.1

*Proof.* We start by proving the "$\supseteq$" direction. Let $\mathbb{Q}$ belong to the right-hand side in (5), and we want to prove that $\mathbb{Q} \in \mathbb{B}_{\varepsilon,\rho}(\mathbb{P})$. From the right-hand side in (5), we know that there exists $\widetilde{\mathbb{P}}$ such that $W_\infty(\mathbb{P}, \widetilde{\mathbb{P}}) \leq \varepsilon$ and $\text{TV}(\widetilde{\mathbb{P}}, \mathbb{Q}) \leq \rho$. Using the definition of the $W_\infty$ distance in (1), we note that $W_\infty(\mathbb{P}, \widetilde{\mathbb{P}}) \leq \varepsilon$ is equivalent to

$$\inf_{\gamma \in \Gamma(\mathbb{P}, \widetilde{\mathbb{P}})} \int_{\mathcal{Z} \times \mathcal{Z}} \mathbb{1}\{\|z_1 - z_2\| > \varepsilon\}\mathrm{d}\gamma(z_1, z_2) \leq 0. \tag{16}$$

Now, since $\mathbb{1}\{\|z_1 - z_2\| > \varepsilon\}$ is a lower semicontinuous function, by [39][Theorem 4.1] we know that there exists a coupling $\gamma_{12}^\star \in \Gamma(\mathbb{P}, \widetilde{\mathbb{P}})$ which attains the infimum in (16). Analogously, since $\mathbb{1}\{\|z_1 - z_2\| > 0\}$ is lower semicontinuous, the same result ensures that there exists a coupling $\gamma_{23}^\star \in \Gamma(\widetilde{\mathbb{P}}, \mathbb{Q})$ which attains the infimum in $\text{TV}(\widetilde{\mathbb{P}}, \mathbb{Q}) \leq \rho$. Since

$$(\pi_2)_{\#}\gamma_{12}^\star = (\pi_1)_{\#}\gamma_{23}^\star = \widetilde{\mathbb{P}},$$

where $\pi_1 : \mathcal{Z}_1 \times \mathcal{Z}_2 \to \mathcal{Z}_1$ and $\pi_2 : \mathcal{Z}_1 \times \mathcal{Z}_2 \to \mathcal{Z}_2$ are the canonical projections, the Gluing lemma [39][pp. 11–12] guarantees that there exists a distribution $\gamma_{123} \in \mathcal{P}(\mathcal{Z} \times \mathcal{Z} \times \mathcal{Z})$ such that $(\pi_{12})_{\#}\gamma_{123} = \gamma_{12}^\star$ and $(\pi_{23})_{\#}\gamma_{123} = \gamma_{23}^\star$. We now construct $\gamma_{13} := (\pi_{13})_{\#}\gamma_{123}$, which

can be easily shown to be a coupling of $\mathbb{P}$ and $\mathbb{Q}$. Then, we have that

$$\int_{\mathcal{Z}\times\mathcal{Z}} \mathbb{1}\{\|z_1-z_3\| > \varepsilon\}\mathrm{d}\gamma_{13}(z_1,z_3) = \int_{\mathcal{Z}\times\mathcal{Z}\times\mathcal{Z}} \mathbb{1}\{\|z_1-z_3\| > \varepsilon\}\mathrm{d}\gamma_{123}(z_1,z_2,z_3)$$

$$= \int_{\mathcal{Z}\times\mathcal{Z}\times\mathcal{Z}} \mathbb{1}\{\|z_1-z_2+z_2-z_3\| > \varepsilon\}\mathrm{d}\gamma_{123}(z_1,z_2,z_3)$$

$$\leq \int_{\mathcal{Z}\times\mathcal{Z}\times\mathcal{Z}} \mathbb{1}\{\|z_1-z_2\|+\|z_2-z_3\| > \varepsilon\}\mathrm{d}\gamma_{123}(z_1,z_2,z_3)$$

$$\leq \int_{\mathcal{Z}\times\mathcal{Z}\times\mathcal{Z}} \left(\mathbb{1}\{\|z_1-z_2\| > \varepsilon\} + \mathbb{1}\{\|z_2-z_3\| > 0\}\right)\mathrm{d}\gamma_{123}(z_1,z_2,z_3)$$

$$= \int_{\mathcal{Z}\times\mathcal{Z}} \mathbb{1}\{\|z_1-z_2\| > \varepsilon\}\mathrm{d}\gamma_{12}^{\star}(z_1,z_2) + \int_{\mathcal{Z}\times\mathcal{Z}} \mathbb{1}\{\|z_2-z_3\| > 0\}\mathrm{d}\gamma_{23}^{\star}(z_2,z_3)$$

$$\leq 0 + \rho = \rho,$$

where the first inequality is a consequence of the triangle inequality, and the second inequality follows by noticing that the event $\{\|z_1-z_2\|+\|z_2-z_3\| > \varepsilon\}$ is contained in $\{\|z_1-z_2\| > \varepsilon\} \cup \{\|z_2-z_3\| > 0\}$. Therefore,

$$\inf_{\gamma\in\Gamma(\mathbb{P},\mathbb{Q})}\int_{\mathcal{Z}\times\mathcal{Z}} \mathbb{1}\{\|z_1-z_3\| > \varepsilon\}\mathrm{d}\gamma(z_1,z_3) \leq \rho,$$

showing that $\mathrm{LP}_{\varepsilon}(\mathbb{P},\mathbb{Q}) \leq \rho$, and therefore $\mathbb{Q}\in\mathbb{B}_{\varepsilon,\rho}(\mathbb{P})$.

We now prove the "$\subseteq$" direction. Let $\mathbb{Q}\in\mathbb{B}_{\varepsilon,\rho}(\mathbb{P})$. In what follows, we will construct a distribution $\widetilde{\mathbb{P}}$ such that $\mathrm{W}_{\infty}(\mathbb{P},\widetilde{\mathbb{P}}) \leq \varepsilon$ and $\mathrm{TV}(\widetilde{\mathbb{P}},\mathbb{Q}) \leq \rho$, showing that $\mathbb{Q}$ belongs to the right-hand side in (5). Since $\mathbb{1}\{\|z_1-z_2\| > \varepsilon\}$ is lower semicontinuous, again by [39][Theorem 4.1], we know that there exists a coupling $\gamma^{\star} \in \Gamma(\mathbb{P},\mathbb{Q})$ which attains the infimum in $\mathrm{LP}_{\varepsilon}(\mathbb{P},\mathbb{Q}) \leq \rho$. Therefore, $\gamma^{\star}(\|z_1-z_2\| > \varepsilon) = \bar{\rho}$ and $\gamma^{\star}(\|z_1-z_2\| \leq \varepsilon) = 1-\bar{\rho}$, for some $\bar{\rho} \leq \rho$. We define the event $\mathcal{A} := \{\|z_1-z_2\| \leq \varepsilon\}$, and its complement $\mathcal{A}^c = \{\|z_1-z_2\| > \varepsilon\}$, and denote by $\gamma^{\star}|_{\mathcal{A}}$ and $\gamma^{\star}|_{\mathcal{A}^c}$ the restrictions of the distribution $\gamma^{\star}$ to $\mathcal{A}$ and $\mathcal{A}^c$, respectively. We now construct the distribution $\widetilde{\mathbb{P}}$ as follows

$$\widetilde{\mathbb{P}} := (\pi_1)_{\#}\gamma^{\star}|_{\mathcal{A}^c} + (\pi_2)_{\#}\gamma^{\star}|_{\mathcal{A}}.$$

note that $\widetilde{\gamma} = \gamma^*|_{\mathcal{A}} + (\mathrm{Id}\times\mathrm{Id})_{\#}\left((\pi_1)_{\#}\gamma^{\star}|_{\mathcal{A}^c}\right)$ is a coupling between $\mathbb{P}$ and $\widetilde{\mathbb{P}}$. Then, we immediately have that

$$\inf_{\gamma\in\Gamma(\mathbb{P},\widetilde{\mathbb{P}})}\int_{\mathcal{Z}^{\otimes 2}} \mathbb{1}\{\|z_1-z_2\| > \varepsilon\}\mathrm{d}\gamma(z_1,z_2) \leq \int_{\mathcal{Z}\times\mathcal{Z}} \mathbb{1}\{\|z_1-z_2\| > \varepsilon\}\mathrm{d}\widetilde{\gamma}(z_1,z_2)$$

$$= \int_{\mathcal{Z}\times\mathcal{Z}} \mathbb{1}\{\|z_1-z_2\| > \varepsilon\}\mathrm{d}\gamma^*|_{\mathcal{A}}(z_1,z_2)$$

$$+ \int_{\mathcal{Z}\times\mathcal{Z}} \mathbb{1}\{\|z_1-z_2\| > \varepsilon\}\mathrm{d}(\mathrm{Id}\times\mathrm{Id})_{\#}\left((\pi_1)_{\#}\gamma^{\star}|_{\mathcal{A}^c}\right)(z_1,z_2)$$

which is clearly equal to zero, showing that $\mathrm{W}_{\infty}(\mathbb{P},\widetilde{\mathbb{P}}) \leq \varepsilon$. Moreover,

$$\mathrm{TV}(\widetilde{\mathbb{P}},\mathbb{Q}) = \mathrm{TV}\left((\pi_1)_{\#}\gamma^{\star}|_{\mathcal{A}^c} + (\pi_2)_{\#}\gamma^{\star}|_{\mathcal{A}}, (\pi_2)_{\#}\gamma^{\star}|_{\mathcal{A}^c} + (\pi_2)_{\#}\gamma^{\star}|_{\mathcal{A}}\right)$$

$$= \inf_{\gamma\in\Gamma\left((\pi_1)_{\#}\gamma^{\star}|_{\mathcal{A}^c}+(\pi_2)_{\#}\gamma^{\star}|_{\mathcal{A}},\,(\pi_2)_{\#}\gamma^{\star}|_{\mathcal{A}}+(\pi_2)_{\#}\gamma^{\star}|_{\mathcal{A}^c}\right)}\int_{\mathcal{Z}\times\mathcal{Z}} \mathbb{1}\{\|z_1-z_2\| > 0\}\mathrm{d}\gamma(z_1,z_2)$$

$$\leq \inf_{\widehat{\gamma}\in\Gamma\left(\frac{1}{\bar{\rho}}(\pi_1)_{\#}\gamma^{\star}|_{\mathcal{A}^c},\,\frac{1}{\bar{\rho}}(\pi_2)_{\#}\gamma^{\star}|_{\mathcal{A}^c}\right)}\int_{\mathcal{Z}\times\mathcal{Z}} \mathbb{1}\{\|z_1-z_2\| > 0\}\mathrm{d}\left(\bar{\rho}\widehat{\gamma} + (\mathrm{Id}\times\mathrm{Id})_{\#}\left((\pi_2)_{\#}\gamma^{\star}|_{\mathcal{A}}\right)\right)(z_1,z_2)$$

$$= \inf_{\widehat{\gamma}\in\Gamma\left(\frac{1}{\bar{\rho}}(\pi_1)_{\#}\gamma^{\star}|_{\mathcal{A}^c},\,\frac{1}{\bar{\rho}}(\pi_2)_{\#}\gamma^{\star}|_{\mathcal{A}^c}\right)}\int_{\mathcal{Z}\times\mathcal{Z}} \mathbb{1}\{\|z_1-z_2\| > 0\}\mathrm{d}(\bar{\rho}\widehat{\gamma})(z_1,z_2)$$

$$= \bar{\rho}\left(\inf_{\widehat{\gamma}\in\Gamma\left(\frac{1}{\bar{\rho}}(\pi_1)_{\#}\gamma^{\star}|_{\mathcal{A}^c},\,\frac{1}{\bar{\rho}}(\pi_2)_{\#}\gamma^{\star}|_{\mathcal{A}^c}\right)}\int_{\mathcal{Z}\times\mathcal{Z}} \mathbb{1}\{\|z_1-z_2\| > 0\}\mathrm{d}\widehat{\gamma}(z_1,z_2)\right)$$

$$= \bar{\rho}.$$

Here, the first inequality holds since $\bar{\rho}\widehat{\gamma} + (\mathrm{Id} \times \mathrm{Id})_\# ((\pi_2)_\#\gamma^\star|_{\mathcal{A}})$, with $\widehat{\gamma} \in \Gamma\left(\frac{1}{\bar{\rho}}(\pi_1)_\#\gamma^\star|_{\mathcal{A}^c}, \frac{1}{\bar{\rho}}(\pi_2)_\#\gamma^\star|_{\mathcal{A}^c}\right)$, is a coupling of $(\pi_1)_\#\gamma^\star|_{\mathcal{A}^c} + (\pi_2)_\#\gamma^\star|_{\mathcal{A}}$ and $(\pi_2)_\#\gamma^\star|_{\mathcal{A}} + (\pi_2)_\#\gamma^\star|_{\mathcal{A}^c}$. Moreover, the third equality follows from the fact that

$$\int_{\mathcal{Z}\times\mathcal{Z}} \mathbb{1}\{\|z_1 - z_2\| > 0\}\mathrm{d}\left((\mathrm{Id}\times\mathrm{Id})_\# ((\pi_2)_\#\gamma^\star|_{\mathcal{A}})\right)(z_1, z_2) = 0.$$

Finally, the last equality follows from the fact that $\mathcal{A}^c = \{\|z_1 - z_2\| > \varepsilon\}$. This shows that $\mathrm{TV}(\widetilde{\mathbb{P}}, \mathbb{Q}) \leq \rho$, and concludes the proof. $\qquad\square$

### D.1.2 Proof of Corollary 2.2

*Proof.* Assertion (i) follows from (5) by setting $\varepsilon$ to zero, resulting in $\widetilde{\mathbb{P}} = \mathbb{P}$. Moreover, assertion (ii) follows from (5) by setting $\rho = 0$, resulting in $\widetilde{\mathbb{P}} = \mathbb{Q}$. $\qquad\square$

### D.1.3 Proof of Proposition 2.3

*Proof.* We first prove that any distribution $\mathbb{Q} \in \mathbb{B}_{\varepsilon,\rho}(\mathbb{P})$ admits a random variable decomposition $Z_2$ as described in (6). Since $\mathbb{1}\{\|z_1 - z_2\| > \varepsilon\}$ is lower semicontinuous, by [39][Theorem 4.1] there exists a coupling $\gamma^\star \in \Gamma(\mathbb{P}, \mathbb{Q})$ which attains the infimum in $\mathrm{LP}_\varepsilon(\mathbb{P}, \mathbb{Q}) \leq \rho$. Furthermore, given $Z_1 \sim \mathbb{P}$, consider the conditional distribution $Z_2|Z_1 \sim \gamma^\star_{Z_1}$, and define the (random) event $\mathcal{A}_{Z_1} := \{\|z_2 - Z_1\| \leq \epsilon\}$. Moreover, we denote by $\gamma^\star_{Z_1}|_{\mathcal{A}_{Z_1}}$ the restriction of $\gamma^\star_{Z_1}$ to the event $\mathcal{A}_{Z_1}$, and by $\overline{\gamma^\star_{Z_1}|_{\mathcal{A}_{Z_1}}}$ its normalized version. Similarly, $\overline{\gamma^\star_{Z_1}|_{\mathcal{A}^c_{Z_1}}}$ is the normalized version of the restriction to the complement $\mathcal{A}^c_{Z_1}$. We then construct the random variables $B$, $N$, and $C$ as follows:

$$\begin{aligned}
B|Z_1 &\sim \mathrm{Bern}\left(\gamma^\star_{Z_1}\left(\|z_2 - Z_1\| > \varepsilon\right)\right), \\
N|Z_1 &= \mathbb{1}(B = 1) \cdot 0 + \mathbb{1}(B = 0) \cdot (Z_2' - Z_1)|Z_1, \text{and} \\
C|Z_1 &= \mathbb{1}(B = 1) \cdot Z_2''|Z_1 + \mathbb{1}(B = 0) \cdot 0,
\end{aligned} \tag{17}$$

where $Z_2'|Z_1$ and $Z_2''|Z_1$ follow the probability distributions $\overline{\gamma^\star_{Z_1}|_{\mathcal{A}_{Z_1}}}$ and $\overline{\gamma^\star_{Z_1}|_{\mathcal{A}^c_{Z_1}}}$, respectively. Here $B$, $N$, $C$ are dependent with marginals satisfying the properties in the statement of the proposition. We now define $\widetilde{Z}_2 := (Z_1 + N)\mathbb{1}\{B = 0\} + C\mathbb{1}\{B = 1\}$, and aim to show that $Z_2 \overset{d}{=} \widetilde{Z}_2$. Following the construction in (17), conditioning $\widetilde{Z}_2$ on $Z_1$ yields

$$\begin{aligned}
\widetilde{Z}_2|Z_1 &= (Z_1 + N|Z_1) \cdot \mathbb{1}\{B|Z_1 = 0\} + C|Z_1 \cdot \mathbb{1}\{B|Z_1 = 1\} \\
&= Z_2'|Z_1 \cdot \mathbb{1}\{B|Z_1 = 0\} + Z_2''|Z_1 \cdot \mathbb{1}(B|Z_1 = 1).
\end{aligned}$$

Now recall from (17) that the conditional random variable $B|Z_1$ follows a Bernoulli distribution with parameter $\gamma^\star_{Z_1}\left(\|z_2 - Z_1\| > \varepsilon\right) = \gamma^\star_{Z_1}(\mathcal{A}^c_{Z_1})$. Thus, the distribution of $\widetilde{Z}_2|Z_1$ becomes $\gamma^\star_{Z_1}(\mathcal{A}_{Z_1}) \cdot \overline{\gamma^\star_{Z_1}|_{\mathcal{A}_{Z_1}}} + \gamma^\star_{Z_1}(\mathcal{A}^c_{Z_1}) \cdot \overline{\gamma^\star_{Z_1}|_{\mathcal{A}^c_{Z_1}}}$. Moreover, since $\gamma^\star_{Z_1}(\mathcal{A}_{Z_1}) \cdot \overline{\gamma^\star_{Z_1}|_{\mathcal{A}_{Z_1}}} = \gamma^\star_{Z_1}|_{\mathcal{A}_{Z_1}}$ and $\gamma^\star_{Z_1}(\mathcal{A}^c_{Z_1}) \cdot \overline{\gamma^\star_{Z_1}|_{\mathcal{A}^c_{Z_1}}} = \gamma^\star_{Z_1}|_{\mathcal{A}^c_{Z_1}}$, we have that $\widetilde{Z}_2|Z_1 \sim \gamma^\star_{Z_1}$. Therefore, the distribution of $\widetilde{Z}_2$ is is equal to

$$\widetilde{Z}_2 = \mathbb{E}_{Z_1}[\widetilde{Z}_2|Z_1] \sim (\pi_2)_\#\gamma^\star = \mathbb{Q},$$

which concludes the proof of the first direction.

We now prove the converse: any random variable $Z_2$ of the form (6) is distributed according to some distribution $\mathbb{Q}$ belonging to the LP ambiguity set $\mathbb{B}_{\varepsilon,\rho}(\mathbb{P})$. To show this, we employ Proposition 2.1, which reduces the problem to showing that $\mathbb{Q}$ belongs to the union on the right-hand side in (5). We start by defining the random variable

$$Z_3 := (Z_1 + N)\mathbb{1}\{B = 0\} + Z_1\mathbb{1}\{B = 1\},$$

where $Z_1, N$, and $B$ are the same random variables as in the definition of $Z_2$ in (6). Let $\widetilde{\mathbb{P}}$ denote the distribution of $Z_3$. Then, the pair $(Z_1, Z_3)$ induces a coupling $\gamma_{13} \in \Gamma(\mathbb{P}, \widetilde{\mathbb{P}})$. By construction we have $\gamma_{13}(\|z_1 - z_3\| > \varepsilon) = 0$, implying that

$$\inf_{\gamma\in\Gamma(\mathbb{P},\widetilde{\mathbb{P}})} \int_{\mathcal{Z}\times\mathcal{Z}} \mathbb{1}\{\|z_1 - z_3\| > \varepsilon\}\mathrm{d}\gamma(z_1, z_3) \leq \int_{\mathcal{Z}\times\mathcal{Z}} \mathbb{1}\{\|z_1 - z_3\| > \varepsilon\}\mathrm{d}\gamma_{13}(z_1, z_3) \leq 0.$$

Using the definition of the $W_\infty$ distance in (1), this is equivalent to $W_\infty(\mathbb{P}, \widetilde{\mathbb{P}}) \leq \varepsilon$. Next, we verify that $TV(\widetilde{\mathbb{P}}, \mathbb{Q}) \leq \rho$. Note that $(Z_3, Z_2)$ induces a coupling $\gamma_{32} \in \Gamma(\widetilde{\mathbb{P}}, \mathbb{Q})$ satisfying

$$\int_{\mathcal{Z} \times \mathcal{Z}} \mathbb{1}\{\|z_3 - z_2\| > 0\} \mathrm{d}\gamma_{32}(z_3, z_2) \leq 0 \cdot \mathrm{Prob}(B = 0) + 1 \cdot \mathrm{Prob}(B = 1) \leq \rho,$$

where the equality follows from $\|(Z_3 - Z_2)|(B = 0)\| = 0$ and the fact that the indicator function is bounded by 1. Therefore,

$$TV(\widetilde{\mathbb{P}}, \mathbb{Q}) := \inf_{\gamma \in \Gamma(\widetilde{\mathbb{P}}, \mathbb{Q})} \int_{\mathcal{Z} \times \mathcal{Z}} \mathbb{1}\{\|z_1 - z_2\| > 0\} \mathrm{d}\gamma(z_1, z_2) \leq \rho,$$

Putting everything together, we have that $\mathbb{Q} \in \bigcup_{\widetilde{\mathbb{P}}: W_\infty(\mathbb{P}, \widetilde{\mathbb{P}}) \leq \varepsilon} \left\{ \mathbb{Q} \in P(\mathcal{Z}) : TV(\widetilde{\mathbb{P}}, \mathbb{Q}) \leq \rho \right\}$, which completes the proof. $\square$

### D.1.4 Proof of Proposition 2.5

*Proof.* Let $\mathbb{Q} \in \mathbb{B}_{\varepsilon, \rho}(\mathbb{P})$. We will show that $s_\# \mathbb{Q}$ belongs to the LP ambiguity set $\mathbb{B}_{k\varepsilon, \rho}(s_\# \mathbb{P})$.

$$\begin{aligned}
LP_{k\varepsilon}(s_\# \mathbb{P}, s_\# \mathbb{Q}) &= \inf_{\tilde{\gamma} \in \Gamma(s_\# \mathbb{P}, s_\# \mathbb{Q})} \int_{\mathbb{R} \times \mathbb{R}} \mathbb{1}\{|\tilde{z}_1 - \tilde{z}_2| > k\varepsilon\} \mathrm{d}\tilde{\gamma}(\tilde{z}_1, \tilde{z}_2) \\
&= \inf_{\tilde{\gamma} \in (s \times s)_\# \Gamma(\mathbb{P}, \mathbb{Q})} \int_{\mathbb{R} \times \mathbb{R}} \mathbb{1}\{|\tilde{z}_1 - \tilde{z}_2| > k\varepsilon\} \mathrm{d}\tilde{\gamma}(\tilde{z}_1, \tilde{z}_2) \\
&= \inf_{\gamma \in \Gamma(\mathbb{P}, \mathbb{Q})} \int_{\mathbb{R} \times \mathbb{R}} \mathbb{1}\{|\tilde{z}_1 - \tilde{z}_2| > k\varepsilon\} \mathrm{d}((s \times s)_\# \gamma)(\tilde{z}_1, \tilde{z}_2) \\
&= \inf_{\gamma \in \Gamma(\mathbb{P}, \mathbb{Q})} \int_{\mathcal{Z} \times \mathcal{Z}} \mathbb{1}(|s(z_1) - s(z_2)| > k\varepsilon) \mathrm{d}\gamma(z_1, z_2) \\
&\leq \inf_{\gamma \in \Gamma(\mathbb{P}, \mathbb{Q})} \int_{\mathcal{Z} \times \mathcal{Z}} \mathbb{1}(\|z_1 - z_2\| > \varepsilon) \mathrm{d}\gamma(z_1, z_2) \\
&= LP_\varepsilon(\mathbb{P}, \mathbb{Q}),
\end{aligned}$$

where the second equality follows from the equality $\Gamma(s_\# \mathbb{P}, s_\# \mathbb{Q}) = (s \times s)_\# \Gamma(\mathbb{P}, \mathbb{Q})$ (see [3][Lemma 2]), and the inequality follows from the fact that $s$ is $k$-Lipschitz, i.e., $|s(z_1) - s(z_2)| \leq k\|z_1 - z_2\|$. $\square$

## D.2 Proofs of Section 3

### D.2.1 Proof of Proposition 3.4

*Proof.* We prove the proposition in two steps. First, we show that the right-hand side in (10) is an upper bound on the $\beta$-quantile of any distribution in $\mathbb{B}_{\varepsilon, \rho}(\mathbb{P})$. Second, we prove that there exists a sequence of distributions $\mathbb{Q}_n \in \mathbb{B}_{\varepsilon, \rho}(\mathbb{P})$, whose $\beta$-quantiles converge to it.

*Step 1.* We prove, by contradiction, that $\mathrm{Quant}(\beta; \mathbb{Q}) \leq \mathrm{Quant}(\beta + \rho; \mathbb{P}) + \varepsilon$, for all $\mathbb{Q} \in \mathbb{B}_{\varepsilon, \rho}(\mathbb{P})$. Suppose there exists $\widetilde{\mathbb{Q}}$ satisfying

$$\widetilde{\mathbb{Q}} \in \mathbb{B}_{\varepsilon, \rho}(\mathbb{P}) \quad \text{and} \quad \mathrm{Quant}(\beta; \widetilde{\mathbb{Q}}) > \mathrm{Quant}(\beta + \rho; \mathbb{P}) + \varepsilon. \tag{18}$$

We will show that this leads to

$$LP_\varepsilon(\mathbb{P}, \widetilde{\mathbb{Q}}) = \inf_{\gamma \in \Gamma(\mathbb{P}, \widetilde{\mathbb{Q}})} \int_{\mathbb{R} \times \mathbb{R}} \mathbb{1}\{|z_1 - z_2| > \varepsilon\} \mathrm{d}\gamma(z_1, z_2) > \rho.$$

To simplify notation, we define $a := \mathrm{Quant}(\beta + \rho; \mathbb{P})$ and $b := \mathrm{Quant}(\beta + \rho; \mathbb{P}) + \varepsilon$. Following (18), $b$ must satisfy $F_{\widetilde{\mathbb{Q}}}(b) < \beta$. Hence, there exists $\Delta > 0$ such that

$$F_\mathbb{P}(a) - F_{\widetilde{\mathbb{Q}}}(b) \geq \rho + \Delta.$$

Now, for an arbitrary coupling $\gamma \in \Gamma(\mathbb{P}, \widetilde{\mathbb{Q}})$, we have

$$F_{\mathbb{P}}(a) - F_{\widetilde{\mathbb{Q}}}(b)$$

$$= \int_{-\infty}^{a} \int_{-\infty}^{\infty} \mathrm{d}\gamma(z_1, z_2) - \int_{-\infty}^{\infty} \int_{-\infty}^{b} d\gamma(z_1, z_2)$$

$$= \int_{-\infty}^{a} \int_{-\infty}^{b} d\gamma(z_1, z_2) + \int_{-\infty}^{a} \int_{b+}^{\infty} d\gamma(z_1, z_2) - \int_{-\infty}^{a} \int_{-\infty}^{b} \mathrm{d}\gamma(z_1, z_2) - \int_{a+}^{\infty} \int_{-\infty}^{b} d\gamma(z_1, z_2)$$

$$\leq \int_{-\infty}^{a} \int_{b+}^{\infty} \mathbb{1}_{\{|z_1 - z_2| > \varepsilon\}} \mathrm{d}\gamma(z_1, z_2)$$

$$\leq \int_{-\infty}^{\infty} \int_{-\infty}^{\infty} \mathbb{1}_{\{|z_1 - z_2| > \varepsilon\}} \mathrm{d}\gamma(z_1, z_2),$$

Since the above holds for every $\gamma \in \Gamma(\mathbb{P}, \widetilde{\mathbb{Q}})$, we conclude that

$$\inf_{\gamma \in \Gamma(\mathbb{P}, \widetilde{\mathbb{Q}})} \int_{\mathbb{R} \times \mathbb{R}} \mathbb{1}_{\{|z_1 - z_2| > \varepsilon\}} \mathrm{d}\gamma(z_1, z_2) \geq \rho + \Delta,$$

which contradicts the fact that $\widetilde{\mathbb{Q}} \in \mathbb{B}_{\varepsilon, \rho}(\mathbb{P})$. This proves that $\mathrm{Quant}(\beta; \mathbb{Q}) \leq \mathrm{Quant}(\beta + \rho; \mathbb{P}) + \varepsilon$, for all $\mathbb{Q} \in \mathbb{B}_{\varepsilon, \rho}(\mathbb{P})$.

*Step 2.* We construct a sequence of distributions $\mathbb{Q}_n \in \mathbb{B}_{\varepsilon, \rho}(\mathbb{P})$ satisfying, as $n \to \infty$,

$$\mathrm{Quant}(\beta; \mathbb{Q}_n) \to \mathrm{Quant}(\beta + \rho; \mathbb{P}) + \varepsilon.$$

We define the sequence of distributions $\mathbb{Q}_n$ through their cumulative distribution functions as

$$F_{\mathbb{Q}_n}(q) = \begin{cases} F_{\mathbb{P}}(q - \varepsilon), & q < \mathrm{Quant}\left(\beta - \frac{1}{n}; \mathbb{P}\right) + \varepsilon \\ \beta - \frac{1}{n}, & \mathrm{Quant}\left(\beta - \frac{1}{n}; \mathbb{P}\right) + \varepsilon \leq q < \mathrm{Quant}\left(\beta - \frac{1}{n} + \rho; \mathbb{P}\right) + \varepsilon \\ F_{\mathbb{P}}(q - \varepsilon), & q \geq \mathrm{Quant}\left(\beta - \frac{1}{n} + \rho; \mathbb{P}\right) + \varepsilon. \end{cases} \tag{19}$$

To simplify notation, for the rest of the proof, we define $q_n^{(1)} := \mathrm{Quant}(\beta - \frac{1}{n}; \mathbb{P}) + \varepsilon$ and $q_n^{(2)} := \mathrm{Quant}(\beta - \frac{1}{n} + \rho; \mathbb{P}) + \varepsilon$. The intuition behind the construction of $\mathbb{Q}_n$ is as follows: first, $\mathbb{Q}_n$ is obtained by translating the distribution $\mathbb{P}$ to the right by $\varepsilon$, and then, the mass between $[q_n^{(1)}, q_n^{(2)})$ is moved to the point $q_n^{(2)}$. We refer to the illustration on the left in Figure 1 for a visualization of this intuition. From this construction, it is clear that the $\mathrm{LP}_\varepsilon(\mathbb{P}, \mathbb{Q}_n)$ is bounded by

$$F_{\mathbb{Q}_n}\left(q_n^{(2)}\right) - F_{\mathbb{Q}_n}\left(q_n^{(1)}\right) = F_{\mathbb{Q}_n}\left(\mathrm{Quant}\left(\beta - \frac{1}{n} + \rho; \mathbb{P}\right) + \varepsilon\right) - F_{\mathbb{Q}_n}\left(\mathrm{Quant}\left(\beta - \frac{1}{n}; \mathbb{P}\right) + \varepsilon\right)$$

$$= F_{\mathbb{P}}\left(\mathrm{Quant}\left(\beta - \frac{1}{n} + \rho; \mathbb{P}\right)\right) - \left(\beta - \frac{1}{n}\right) = \rho,$$

showing that the sequence $\mathbb{Q}_n$ belongs to the LP ambiguity set $\mathbb{B}_{\varepsilon, \rho}(\mathbb{P})$. Finally, we prove that the sequence of $\beta$-quantiles of $\mathbb{Q}_n$ converges to $\mathrm{Quant}(\beta + \rho; \mathbb{P}) + \varepsilon$ from below. From the construction in (19), we know that the following two properties hold:

- $F_{\mathbb{Q}_n}(q) < \beta, \qquad \forall \, q < q_n^{(2)};$

- $F_{\mathbb{Q}_n}(q) \geq \beta, \qquad \forall \, q \geq q_n^{(2)}, \quad n \geq \frac{1}{\rho}.$

Combining these two inequalities, we have that $\mathrm{Quant}(\beta; \mathbb{Q}_n) = q_n^{(2)}$, which admits a limit as $n$ goes to infinity:

$$q_n^{(2)} = \mathrm{Quant}\left(\beta - \frac{1}{n} + \rho; \mathbb{P}\right) + \varepsilon \xrightarrow{n \to \infty} \mathrm{Quant}(\beta + \rho; \mathbb{P}) + \epsilon$$

where the convergence follows from the left-continuity of the quantile function, which follows from the right-continuity of the cumulative distribution function. This concludes the proof. $\qquad \square$

### D.2.2    Proof of Proposition 3.5

*Proof.* Similarly to Proposition 3.4, we prove this in two steps. First, we show that the right-hand side in (11) is a lower bound on the coverage at $q$ of any distribution in $\mathbb{B}_{\varepsilon,\rho}(\mathbb{P})$. Second, we prove that there exists a sequence of distributions $\mathbb{Q}_n \in \mathbb{B}_{\varepsilon,\rho}(\mathbb{P})$, whose coverage at $q$ converges to it.

*Step 1.* We prove, by contradiction, that $F_{\mathbb{Q}}(q) \geq F_{\mathbb{P}}(q - \varepsilon) - \rho$, for all $\mathbb{Q} \in \mathbb{B}_{\varepsilon,\rho}(\mathbb{P})$. Suppose there exists $\widetilde{\mathbb{Q}}$ satisfying

$$\widetilde{\mathbb{Q}} \in \mathbb{B}_{\varepsilon,\rho}(\mathbb{P}) \quad \text{and} \quad F_{\widetilde{\mathbb{Q}}}(q) < F_{\mathbb{P}}(q - \varepsilon) - \rho. \tag{20}$$

We will show that this leads to

$$\mathrm{LP}_{\varepsilon}(\mathbb{P}, \widetilde{\mathbb{Q}}) = \inf_{\gamma \in \Gamma(\mathbb{P}, \widetilde{\mathbb{Q}})} \int_{\mathbb{R} \times \mathbb{R}} \mathbb{1}\{|z_1 - z_2| > \varepsilon\} d\gamma(z_1, z_2) > \rho.$$

From the inequality in (20), we know that there exists $\Delta > 0$ such that

$$F_{\widetilde{\mathbb{Q}}}(q) \leq F_{\mathbb{P}}(q - \varepsilon) - (\rho + \Delta).$$

Meanwhile, for any coupling $\gamma \in \Gamma(\mathbb{P}, \widetilde{\mathbb{Q}})$, we have

$\rho + \Delta \leq F_{\mathbb{P}}(q - \varepsilon) - F_{\widetilde{\mathbb{Q}}}(q)$

$$= \int_{-\infty}^{q-\varepsilon} \int_{-\infty}^{\infty} d\gamma(z_1, z_2) - \int_{-\infty}^{\infty} \int_{-\infty}^{q} d\gamma(z_1, z_2)$$

$$= \int_{-\infty}^{q-\varepsilon} \int_{-\infty}^{q} d\gamma(z_1, z_2) + \int_{-\infty}^{q-\varepsilon} \int_{q^+}^{\infty} d\gamma(z_1, z_2) - \int_{-\infty}^{q-\varepsilon} \int_{-\infty}^{q} d\gamma(z_1, z_2) - \int_{(q-\varepsilon)^+}^{\infty} \int_{-\infty}^{q} d\gamma(z_1, z_2)$$

$$\leq \int_{-\infty}^{q-\varepsilon} \int_{q^+}^{\infty} \mathbb{1}_{\{|z_1 - z_2| > \varepsilon\}} \, d\gamma(z_1, z_2)$$

$$\leq \int_{\mathbb{R} \times \mathbb{R}} \mathbb{1}_{\{|z_1 - z_2| > \varepsilon\}} \, d\gamma(z_1, z_2).$$

Taking an infimum over $\gamma \in \Gamma(\mathbb{P}, \widetilde{\mathbb{Q}})$, we obtain that the $\mathrm{LP}_{\varepsilon}(\mathbb{P}, \widetilde{\mathbb{Q}}) > \rho$, which contradicts the fact that $\widetilde{\mathbb{Q}} \in \mathbb{B}_{\varepsilon,\rho}(\mathbb{P})$. This proves that $F_{\mathbb{Q}}(q) \geq F_{\mathbb{P}}(q - \varepsilon) - \rho$, for all $\mathbb{Q} \in \mathbb{B}_{\varepsilon,\rho}(\mathbb{P})$.

*Step 2.* We construct a sequence of distributions $\mathbb{Q}_n \in \mathbb{B}_{\varepsilon,\rho}(\mathbb{P})$ satisfying, as $n \to \infty$,

$$F_{\mathbb{Q}_n}(q) \to F_{\mathbb{P}}(q - \varepsilon) - \rho.$$

We define the sequence of distributions $\mathbb{Q}_n$ through their cumulative distribution functions as

$$F_{\mathbb{Q}_n}(\gamma) = \begin{cases} F_{\mathbb{P}}(\gamma - \varepsilon), & \gamma < \mathrm{Quant}\left(F_{\mathbb{P}}(q - \varepsilon) - \rho + \frac{1}{n}; \mathbb{P}\right) + \varepsilon \\ F_{\mathbb{P}}(q - \varepsilon) - \rho + \frac{1}{n}, & \mathrm{Quant}\left(F_{\mathbb{P}}(q - \varepsilon) - \rho + \frac{1}{n}; \mathbb{P}\right) + \varepsilon \leq \gamma \\ & \quad < \mathrm{Quant}\left(F_{\mathbb{P}}(q - \varepsilon) + \frac{1}{n}; \mathbb{P}\right) + \varepsilon \\ F_{\mathbb{P}}(\gamma - \varepsilon), & \gamma \geq \mathrm{Quant}\left(F_{\mathbb{P}}(q - \varepsilon) + \frac{1}{n}; \mathbb{P}\right) + \varepsilon. \end{cases}$$

To simplify notation, for the rest of the proof, we define $q_n^{(1)} = \mathrm{Quant}(F_{\mathbb{P}}(q - \varepsilon) - \rho + \frac{1}{n}; \mathbb{P}) + \varepsilon$ and $q_n^{(2)} = \mathrm{Quant}(F_{\mathbb{P}}(q - \varepsilon) + \frac{1}{n}; \mathbb{P}) + \varepsilon$. The intuition behind the construction of $\mathbb{Q}_n$ is as follows: first, $\mathbb{Q}_n$ is obtained by translating the distribution $\mathbb{P}$ to the right by $\varepsilon$, and then, the mass between $[q_n^{(1)}, q_n^{(2)})$ is moved to the point $q_n^{(2)}$. We refer to the illustration on the right in Figure 1 for a visualization of this intuition. From this construction, it is clear that the $\mathrm{LP}_{\varepsilon}(\mathbb{P}, \mathbb{Q}_n)$ is bounded by

$$F_{\mathbb{Q}_n}\left(q_n^{(2)}\right) - F_{\mathbb{Q}_n}\left(q_n^{(1)}\right)$$

$$= F_{\mathbb{P}}\left(\mathrm{Quant}\left(F_{\mathbb{P}}(q - \varepsilon) + \frac{1}{n}; \mathbb{P}\right)\right) - F_{\mathbb{P}}\left(\mathrm{Quant}\left(F_{\mathbb{P}}(q - \varepsilon) - \rho + \frac{1}{n}; \mathbb{P}\right)\right)$$

$$= F_{\mathbb{P}}(q - \varepsilon) + \frac{1}{n} - \left(F_{\mathbb{P}}(q - \varepsilon) - \rho + \frac{1}{n}\right) = \rho,$$

showing that the sequence $\mathbb{Q}_n$ belongs to the LP ambiguity set $\mathbb{B}_{\varepsilon,\rho}(\mathbb{P})$. Moreover, when $n \geq \frac{1}{\rho}$, we have that $q \in [q_n^{(1)}, q_n^{(2)})$ holds, and therefore

$$F_{\mathbb{Q}_n}(q) = F_{\mathbb{P}}(q - \varepsilon) - \rho + \frac{1}{n} \xrightarrow{n \to \infty} F_{\mathbb{P}}(q - \varepsilon) - \rho.$$

This concludes the proof. □

### D.3 Proofs of Section 4

#### D.3.1 Proof of Theorem 4.1

*Proof.* By conditioning on $\{(X_i, Y_i)\}_{i=1}^n$, we obtain

$$\text{Prob}\left\{Y_{n+1} \in C_{\varepsilon,\rho}(X_{n+1}; \widehat{\mathbb{P}}_n) | \{(X_i, Y_i)\}_{i=1}^n\right\} = F_{\mathbb{P}_{\text{test}}}\left(\text{Quant}_{\varepsilon,\rho}^{\text{WC}}(1 - \alpha; \widehat{\mathbb{P}}_n)\right)$$
$$= F_{\mathbb{P}_{\text{test}}}\left(\text{Quant}(1 - \alpha + \rho; \widehat{\mathbb{P}}_n) + \varepsilon\right)$$
$$\geq F_{\mathbb{P}}\left(\text{Quant}(1 - \alpha + \rho; \widehat{\mathbb{P}}_n) + \varepsilon - \varepsilon\right) - \rho$$
$$= F_{\mathbb{P}}\left(\text{Quant}(1 - \alpha + \rho, \widehat{\mathbb{P}}_n)\right) - \rho,$$

where the first equality follows from Definition 12, the second equality follows from Proposition 3.4, and the first inequality is a consequence of Proposition 3.5. Now, taking the expectation with respect to $\{(X_i, Y_i)\}_{i=1}^n$, we obtain

$$\text{Prob}\left\{Y_{n+1} \in C_{\varepsilon,\rho}^{1-\alpha}\left(X_{n+1}; \widehat{\mathbb{P}}_n\right)\right\} \geq \mathbb{E}\left[F_{\mathbb{P}}\left(\text{Quant}(1 - \alpha + \rho, \widehat{\mathbb{P}}_n)\right)\right] - \rho$$
$$\geq \frac{\lceil n(1 - \alpha + \rho) \rceil}{n + 1} - \rho,$$

where the second inequality follows from the guarantee $\mathbb{E}\left[F_{\mathbb{P}}(\text{Quant}(\beta; \widehat{\mathbb{P}}_n))\right] \geq \lceil n\beta \rceil/(n + 1)$ (see [9][Lemma D.3]). This concludes the proof. □

#### D.3.2 Proof of Corollary 4.2

*Proof.* Note that $\lceil n(1 - \beta + \rho) \rceil/(n + 1) - \rho \geq 1 - \alpha$ is guaranteed by $n(1 - \beta + \rho) \geq (n + 1)(1 - \alpha + \rho) + 1$, which is further guaranteed by $\beta \leq \alpha + (\alpha - \rho - 2)/n$. This concludes the proof. □

