# OpenReview forum: "Conformal Prediction under Lévy-Prokhorov Distribution Shifts: Robustness to Local and Global Perturbations"
_NeurIPS.cc/2025/Conference — NeurIPS 2025 poster_

### Official Review · Reviewer_YGEH · 2025-06-16

**Clarity:** 3
**Significance:** 2
**Originality:** 2
**Rating:** 4
**Confidence:** 4

**Summary:**

This paper proposes a new framework for robust conformal prediction under distribution shift using Lévy–Prokhorov (LP) ambiguity sets, which capture both local and global perturbations. It shows that the LP ambiguity set can be effectively propagated through scoring functions, simplifying high-dimensional distribution shifts into tractable one-dimensional shifts. Closed-form expressions for worst-case quantiles and coverage are derived. The paper also introduces a data-driven method to estimate LP parameters and demonstrates the efficacy of the proposed method on benchmark datasets, including MNIST, ImageNet, and iWildCam.

**Questions:**

- See **Weaknesses**.
- In **Proposition 2.3**, better avoid using $n$ as it stands for sample size later (Section 4) in the paper.
- Second line of **Definition 3.2**: a typo in "$\mathbb{Q}$.or $q\in\mathbb{R}$"?
- In Figures 5 and 6 in the Appendix, why is the lower bound for the candidate values of $\varepsilon$ set at 0.5? I wonder what would happen if $\varepsilon$ is less than 0.5.

**Ethical Concerns:**

["NO or VERY MINOR ethics concerns only"]

**Final Justification:**

Most of my concerns have been addressed and I would like to keep my positive score.

**Limitations:**

Yes.

**Quality:**

3

**Strengths And Weaknesses:**

**Strengths**
- This paper extends existing works using ambiguity sets, including $f$-divergence ambiguity sets.
- The theoretical contributions are significant, with rigorous derivations for worst-case quantiles and coverage under LP ambiguity.


**Weaknesses**
- The connection to the work of Cauchois et al. [9] should be discussed in more depth. While the paper does cite [9] and acknowledges some high-level similarities, the key differences in ideas, methodology, theoretical formulation, and proof techniques are not fully discussed.
- Estimation of the LP ambiguity set parameters $\varepsilon$ and $\rho$ is critical for the practical implementation of the proposed approach, and thus I would suggest moving it to the main paper.
- Although the LP framework is highly general, experiments are limited to classification tasks. Can you add more experiments on regression settings?

---

> ### Author Rebuttal · Authors · 2025-07-30
>
> We are very grateful for your time and effort in reviewing our paper. Thank you for the positive assessment of our work. Below, we first provide answers to your questions and then to the highlighted weaknesses.
>
> ----
>
> **Question 1.** We proceed to answer to the three weaknesses.
>
> *Weakness 1.* We appreciate the suggestion to elaborate on the connection with [9]. Our work was indeed inspired by their formulation (as noted in the introduction) and, in particular, by their treatment of the fact that only samples from the calibration distribution are available. This influence is reflected in Theorem 4.1, whose proof adapts ideas from their Theorem 1 to our setting.
>
> That said, there are key differences in ideas, methodology, theoretical formulation, and proof techniques. In terms of ideas, we replace the $f$‑divergence ambiguity sets in [9] with Lévy–Prokhorov (LP) sets. $f$‑divergences cannot accommodate distribution shifts with disjoint support, which occur in practice under adversarial perturbations, label shifts, or severe data contamination. The LP framework addresses this limitation and offers a natural decomposition into local and global perturbations, enabling separate reasoning about small‑magnitude shifts and large‑magnitude deviations.
>
> Methodologically, our shift model is defined via an optimal transport (OT) discrepancy, placing it in a different mathematical class than $f$‑divergences. OT‑based sets have a distinct geometry and constraint structure, requiring tools from OT theory rather than the convex‑analytic duality used for $f$‑divergences. This also shapes the proof strategy: in Propositions 3.4 and 3.5, we explicitly construct worst‑case distributions within LP sets and derive closed‑form characterizations of worst‑case quantiles and coverage guarantees. These arguments, tailored to the LP geometry, differ fundamentally from the duality‑based derivations in [9] and, we believe, are more broadly applicable to other OT‑driven robustness models.
>
> *Weakness 2.* Thank you for the suggestion. Space permitting, we will try to move as much of the discussion on the estimation of $(\varepsilon,\rho)$ as possible to the main text.
>
> *Weakness 3.* We agree that experiments on regression settings would further strengthen the paper. In response to your suggestion, we are adding experiments on the PovertyMap dataset introduced by Yeh et al. (“Using publicly available satellite imagery and deep learning to understand economic well-being in Africa”). This dataset contains multispectral Landsat imagery paired with a continuous-valued asset wealth index, with natural distribution shifts arising from geographic (cross-country) and urban–rural differences. We have set up access via Google Earth Engine and are awaiting approval; results will be appended once available.
>
> This setting is analogous to iWildCam but with continuous labels, for which we take the residual $s(X_n, Y_n) = |Y_n - f(X_n)|$ as the score. Based on our theoretical framework, we expect LP-Robust Conformal to perform well in this domain.
>
>
>
> -------------------
>
>
>
> **Questions 2+3.** Thank you for the suggestion to use a different letter for the local perturbation in Proposition 2.3. We have double‑checked and confirm that Definition 3.2 is correct as stated and does not contain a typo.
>
>
>
> -------------------
>
>
>
> **Question 4.** Thank you for the question. For smaller values of $\epsilon$, we observe a sharp increase in the corresponding $\rho$. This still yields valid coverage (consistent with the trade‑off described in Remark A.1, where smaller $\epsilon$ typically corresponds to larger $\rho$). However, because $\rho$ shifts the quantile level directly (see Proposition 3.4), even moderate increases in $\rho$ have a stronger effect on coverage than changes in $\epsilon$. As a result, parameter pairs with very small $\epsilon$ and large $\rho$ tend to be less efficient and are not selected under the coverage‑efficiency trade‑off in Corollary 4.2.
>
> In the plots, we truncated the $\epsilon$ search range to $[0.5,\cdot]$ purely for illustration and readability, since smaller values would make $\rho$ spike sharply and obscure the relevant Pareto‑optimal region. In the revised version, we can enlarge the window and include an explanation of this phenomenon for completeness.

---

> > ### Author Response · Authors · 2025-08-04
> > **Additional real-world regression experiment**
> >
> > To provide additional results on real-world distribution shift, particularly for regression tasks, we aimed to include the PovertyMap dataset in our experiments. However, accessing PovertyMap requires an approval process that may take several weeks. To avoid delaying the response, we selected the Shifts Weather Prediction dataset instead, which offers a high-quality regression task under naturally occurring covariate and domain shifts. We believe this dataset serves as a strong and realistic benchmark, and complements our existing evaluation by further demonstrating the robustness and generalizability of our method in real-world settings.
> >
> >
> > **Summary of the dataset and regression task:**
> >
> > The Shifts Weather Prediction dataset defines a scalar regression task: predicting surface temperature from 123 heterogeneous meteorological covariates and 4 metadata attributes (latitude, longitude, time, and climate type). It contains 10 million entries collected from September 2018 to September 2019 across diverse climate zones, reflecting real-world distribution shifts. A canonical split defines in-domain data (Tropical, Dry, Mild Temperate) and out-of-domain test sets (Snow, Polar), enabling robust evaluation under temporal and geographic variation. In our experiments, we predict the temperature using a CatBoostRegressor ensemble with 10 members as the forward model, and use the absolute error between the predicted and true temperature as the conformity score. To assess robustness in this setting, we applied our proposed LP-robust conformal prediction method, alongside five benchmark approaches including standard split conformal prediction and distribution shift–aware baselines. We estimated the $(\varepsilon, \rho)$ parameters in our method using the algorithm described in Appendix A.
> >
> > | Method       | Coverage | Interval Length |
> > |---------------|------------------|-------------------|
> > | SC     | 0.771    | 4.754           |
> > | Weight      | 0.783    | 4.892           |
> > | $\chi^2$          | 0.855    | 5.902           |
> > | RSCP          | 0.902    | 7.272           |
> > | FG-CP  | 0.912    | 7.154           |
> > | $\mathrm{LP}_\mathbf{\varepsilon}$     | **0.900**    | **6.824**           |
> >
> > **Explanation of results:**
> >
> > Classical conformal prediction (SC), covariate shift–aware weighting (Weight), and the $f$-divergence–based method ($\chi^2$) all exhibit undercoverage, as they fail to account for global distribution shifts. Among the methods achieving valid coverage—RSCP, fine-grained CP (FG-CP), and our LP-robust approach—ours yields the smallest prediction intervals. RSCP produces wider sets potentially due to smoothed scores, and FG-CP slightly overcovers due to challenges in estimating conditional divergences. In contrast, our method achieves both valid and sharp coverage, suggesting that the learned $(\varepsilon, \rho)$ pair effectively captures the shift.

---

> > > ### Comment · Reviewer_YGEH · 2025-08-05
> > >
> > > Thanks for the detailed responses, which addressed most of my concerns. I would like to maintain my positive score.

---

### Official Review · Reviewer_Y1BD · 2025-06-23

**Clarity:** 4
**Significance:** 3
**Originality:** 3
**Rating:** 4
**Confidence:** 3

**Summary:**

This paper proposes a robust conformal prediction framework under Lévy-Prokhorov (LP) distribution shifts, supported by rigorous theoretical guarantees and various real-data experiments results.

**Questions:**

1. Beyond the mathematical formulation in Proposition 2.3, how can we intuitively understand the difference between local perturbation and global perturbation?
2. In Figure 1, what do the blue and red lines represent, respectively? The caption should provide more detail to avoid ambiguity.
3. About the experiments results, how to estimate the $\varepsilon$ and $\rho$ in Figure 6?

**Ethical Concerns:**

["NO or VERY MINOR ethics concerns only"]

**Final Justification:**

Thanks for the responses. I keep my positive score for this paper.

**Limitations:**

Yes, limitations are discussed in the Appendix of the paper.

**Paper Formatting Concerns:**

line 335, what is Figure 5.2 indicating for?

**Quality:**

3

**Strengths And Weaknesses:**

Strengths:
- The problem addressed is important and well-motivated.
- The use of the Lévy-Prokhorov pseudo-metric to model distributional shifts is novel and well-defined, with two robustness parameters, local parameter $\varepsilon$ and global parameter $\rho$.
- The authors conduct well-designed experiments to demonstrate the effectiveness of their method.

Weaknesses:
- In practice, the nature of distribution shifts is typically unknown. However, the experimental evaluation is limited to synthetic local and global perturbations and includes only one real-world dataset, without exploring a broader range of naturally occurring distribution shifts. This limitation raises concerns about the practical robustness and generalizability of the proposed method in real-world applications.

---

> ### Author Rebuttal · Authors · 2025-07-30
>
> We are very grateful for your time and effort in reviewing our paper. Thank you for the positive assessment of our work. Below, we first provide answers to your questions and then to the highlighted weaknesses.
>
> ----
>
> **Questions 1+3.** Thank you for these questions. Intuitively, the local perturbation parameter $\epsilon$ captures small, dense, and continuous shifts (such as Gaussian noise, mild input drift, or other shape-preserving distortions), while the global perturbation parameter $\rho$ models sparse or structural changes (such as label flips, adversarial attacks, or support mismatches). This decomposition, formalized in Propositions 2.1 and 2.3, allows the LP-bounded shift model to represent both small-magnitude changes and large, isolated deviations. Many realistic shifts (bounded $\ell_{\infty}$ attacks, mild covariate drift, or mixed noise-plus-outlier scenarios) fall within this class.
>
> In practice, $(\epsilon,\rho)$ can be approximated from a modest number of calibration and test samples using the procedure in Appendix B (Algorithm 1, which we will move to the main text for clarity). The method proceeds in two steps:
>
> (i) grid search over $\epsilon$, and for each candidate $\epsilon$ estimate $\rho$ as the Lévy-Prokhorov (Wasserstein-$\infty$) distance between independent calibration and test score distributions, computed via 1D optimal transport with cost $\mathbb{1}(|x-y| \geq \epsilon)$;
>
> (ii) among candidate pairs, select the one minimizing the worst‑case $(1-\alpha)$-quantile (Corollary 4.2) to balance robustness and prediction-set size. Calibration scores used for parameter estimation are kept independent from those used to compute the conformal quantile to preserve validity.
>
> This process reveals a natural trade-off: smaller $\epsilon$ typically corresponds to larger $\rho$, and vice versa. As shown in Figures 5 and 6, Pareto-optimal pairs (small $\epsilon$ with moderate $\rho$) achieve the best robustness-efficiency balance. The parameter $\epsilon$ affects set size additively, while $\rho$ shifts the quantile level itself (Proposition 3.4), making underestimation of $\rho$ riskier for coverage than underestimating $\epsilon$.
>
> ----
>
> **Question 2.** Thank you for pointing this out. In Figure 1, the red curve represents the nominal score distribution $F_{\mathbb{P}}$, while the blue curve shows an adversarial distribution in the LP ambiguity set $\mathbb{B}_{\epsilon,\rho}(\mathbb{P})$, chosen to either maximize the $\beta$-quantile (left) or minimize coverage at a fixed threshold $q$ (right). The figure illustrates exactly the construction of the worst‑case distribution used in the proofs of Propositions 3.4 and 3.5, where it is cited for intuition. We will clarify both the meaning of the curves and this connection in the caption.
>
> ----
>
> **Weakness.** We agree that broader empirical validation would further strengthen the paper. While synthetic shifts are useful for isolating and visualizing the effects of local and global perturbations, we also evaluate on iWildCam, a benchmark dataset with naturally occurring covariate shift. Our estimated parameters align closely with the Pareto frontier in Figure 4 and yield tight prediction sets, supporting the method’s practical utility.
>
> In response to your suggestion, we are adding experiments on the PovertyMap dataset introduced by Yeh et al. (“Using publicly available satellite imagery and deep learning to understand economic well-being in Africa”). This dataset contains multispectral Landsat imagery paired with a continuous-valued asset wealth index, with natural distribution shifts arising from geographic (cross-country) and urban-rural differences. We have set up access via Google Earth Engine and are awaiting approval; results will be appended once available.
>
> This setting is analogous to iWildCam but with continuous labels, for which we take the residual $s(X_n, Y_n) = |Y_n - f(X_n)|$ as the score. Based on our theoretical framework, we expect LP-Robust Conformal to perform well in this domain.

---

> > ### Author Response · Authors · 2025-08-04
> > **Additional real-world regression experiment**
> >
> > To provide additional results on real-world distribution shift, particularly for regression tasks, we aimed to include the PovertyMap dataset in our experiments. However, accessing PovertyMap requires an approval process that may take several weeks. To avoid delaying the response, we selected the Shifts Weather Prediction dataset instead, which offers a high-quality regression task under naturally occurring covariate and domain shifts. We believe this dataset serves as a strong and realistic benchmark, and complements our existing evaluation by further demonstrating the robustness and generalizability of our method in real-world settings.
> >
> >
> > **Summary of the dataset and regression task:**
> >
> > The Shifts Weather Prediction dataset defines a scalar regression task: predicting surface temperature from 123 heterogeneous meteorological covariates and 4 metadata attributes (latitude, longitude, time, and climate type). It contains 10 million entries collected from September 2018 to September 2019 across diverse climate zones, reflecting real-world distribution shifts. A canonical split defines in-domain data (Tropical, Dry, Mild Temperate) and out-of-domain test sets (Snow, Polar), enabling robust evaluation under temporal and geographic variation. In our experiments, we predict the temperature using a CatBoostRegressor ensemble with 10 members as the forward model, and use the absolute error between the predicted and true temperature as the conformity score. To assess robustness in this setting, we applied our proposed LP-robust conformal prediction method, alongside five benchmark approaches including standard split conformal prediction and distribution shift–aware baselines. We estimated the $(\varepsilon, \rho)$ parameters in our method using the algorithm described in Appendix A.
> >
> > | Method       | Coverage | Interval Length |
> > |---------------|------------------|-------------------|
> > | SC     | 0.771    | 4.754           |
> > | Weight      | 0.783    | 4.892           |
> > | $\chi^2$          | 0.855    | 5.902           |
> > | RSCP          | 0.902    | 7.272           |
> > | FG-CP  | 0.912    | 7.154           |
> > | $\mathrm{LP}_\mathbf{\varepsilon}$     | **0.900**    | **6.824**           |
> >
> > **Explanation of results:**
> >
> > Classical conformal prediction (SC), covariate shift–aware weighting (Weight), and the $f$-divergence–based method ($\chi^2$) all exhibit undercoverage, as they fail to account for global distribution shifts. Among the methods achieving valid coverage—RSCP, fine-grained CP (FG-CP), and our LP-robust approach—ours yields the smallest prediction intervals. RSCP produces wider sets potentially due to smoothed scores, and FG-CP slightly overcovers due to challenges in estimating conditional divergences. In contrast, our method achieves both valid and sharp coverage, suggesting that the learned $(\varepsilon, \rho)$ pair effectively captures the shift.

---

> > > ### Comment · Reviewer_Y1BD · 2025-08-04
> > > **Acknowledgement and response to the rebuttal**
> > >
> > > Thank you for the detailed response and additional work on experiments. I keep my positive score of 4.

---

### Official Review · Reviewer_Ro7g · 2025-06-23

**Clarity:** 3
**Significance:** 3
**Originality:** 3
**Rating:** 4
**Confidence:** 3

**Summary:**

This work studies the problem of distribution shift in conformal prediction and presents a novel framework that leverages the Lévy-Prokhorov (LP) metric to provide robustness guarantees. The authors derive theoretical results and validate the approach through empirical studies involving synthetic perturbations and a real dataset.

**Questions:**

1. Can the authors elaborate on how the robustness parameters are interpreted or estimated from data in a real application setting?
2. The theoretical results in Theorem 4.1 rely on LP-bounded shifts. In practice, how restrictive is this assumption? Is it easy to verify or approximate?

**Ethical Concerns:**

["NO or VERY MINOR ethics concerns only"]

**Final Justification:**

Thank you for your efforts to address my concerns! I maintain my positive score of 4.

**Limitations:**

Yes, the limitations of the paper is provided in Appendix.

**Paper Formatting Concerns:**

No.

**Quality:**

3

**Strengths And Weaknesses:**

Strengths:
- The topic is highly relevant to real-world applications, where test-time distribution shifts are often inevitable.
- The incorporation of the LP metric into conformal prediction is conceptually interesting and mathematically sound. It provides a unified view of local and global shift scenarios.
- The experiments are generally consistent with the theoretical findings and demonstrate the practical utility of the method.

Weaknesses:
- While the theoretical framework accommodates a broad class of shifts, the empirical validation remains limited in scope. Specifically, the experiments mainly consider artificial perturbations, and more extensive evaluation on multiple real-world benchmarks would strengthen the practical relevance of the work.
- The connection between theoretical robustness parameters and their empirical estimation remains unclear, and the method may be difficult to apply in practice without further guidance.

---

> ### Author Rebuttal · Authors · 2025-07-30
>
> We are very grateful for your time and effort in reviewing our paper. Thank you for the positive assessment of our work. Below, we first provide answers to your questions and then to the highlighted weaknesses.
>
> ----
> **Question 1.** Thank you for highlighting this important aspect. Intuitively, the local perturbation parameter $\epsilon$ captures small, dense, and continuous shifts (e.g., Gaussian noise or input drift) that alter the distribution’s shape, while the global perturbation parameter $\rho$ accounts for sparse or structural changes (e.g., label flips, adversarial attacks, or support mismatches). This decomposition is formally described in Propositions 2.1 and 2.3.
>
> For estimation in practice, Appendix B (Algorithm 1) describes an efficient two‑step procedure using a small number of calibration and test samples:
>
> (i) Grid search over $\epsilon$: For each candidate $\epsilon$, estimate $\rho$ as the LP distance between independent calibration and test score distributions, computed via 1D optimal transport with cost $\mathbb{1}(|x - y| \geq \epsilon)$. Each $(\epsilon_i, \rho_i)$ defines a valid LP ambiguity set satisfying empirical coverage.
>
> (ii) Selection for efficiency: Among candidate pairs, choose the one minimizing the worst‑case $(1-\alpha)$-quantile (Corollary 4.2) to balance robustness and set size. Calibration scores for parameter estimation are kept independent from those used to compute the conformal quantile to ensure validity.
>
> This yields a natural trade-off: smaller $\epsilon$ typically corresponds to larger $\rho$, and vice versa. As shown in Figures 5 and 6, Pareto-optimal pairs (small $\epsilon$ with moderate $\rho$) offer the best balance between robustness and efficiency. Importantly, $\epsilon$ contributes additively to the prediction set size, whereas $\rho$ shifts the quantile level itself (see the worst-case quantile characterization in Proposition 3.4) and therefore has a more pronounced effect on coverage. Consequently, underestimating $\rho$ poses a greater risk (potentially violating coverage) while underestimating $\epsilon$ primarily impacts efficiency.
>
> ----
>
> **Question 2.** The LP‑bounded shift model is intentionally designed to be flexible: it only constrains most of the probability mass to remain within a small tolerance while allowing a limited fraction to be displaced arbitrarily. This enables it to capture both smooth, small‑magnitude perturbations and large, sparse deviations, covering a wide range of realistic shifts: from gradual covariate drift to adversarial perturbations and isolated but severe outliers. Many common real‑world shifts (e.g., bounded $\ell_{\infty}$ attacks, mild input drift, or mixed noise‑plus-outlier scenarios) fall within this class.
>
> While exact verification would require the full test distribution, the parameters $(\varepsilon,\rho)$ can be approximated reliably from a modest number of calibration and test samples. In our experiments, we estimate them by comparing the empirical score distributions via a simple 1D optimal transport calculation. Conservative estimates ensure that the method’s coverage guarantee remains valid, and the results show that these estimates are accurate enough to retain efficiency.
>
> We agree that parameter estimation is critical and have developed an explicit procedure (Algorithm 1 in Appendix B). In the revision, we will move this discussion to the main text (Section 4) for greater visibility, add further intuition and empirical support, and release our implementation to facilitate reproducibility.
>
> ----
>
> **Weaknesses.** Answer to Weaknesses: We agree that broader empirical validation would further strengthen the paper. While synthetic shifts are useful for isolating and visualizing the effects of local and global perturbations, we also evaluate on iWildCam, a benchmark dataset with naturally occurring covariate shift. Our estimated parameters align closely with the Pareto frontier in Figure 4 and yield tight prediction sets, supporting the method’s practical utility.
>
> In response to your suggestion, we are adding experiments on the PovertyMap dataset introduced by Yeh et al. (“Using publicly available satellite imagery and deep learning to understand economic well-being in Africa”). This dataset contains multispectral Landsat imagery paired with a continuous-valued asset wealth index, with natural distribution shifts arising from geographic (cross-country) and urban-rural differences. We have set up access via Google Earth Engine and are awaiting approval; results will be appended once available.
>
> This setting is analogous to iWildCam but with continuous labels, for which we take the residual $s(X_n, Y_n) = |Y_n - f(X_n)|$ as the score. Based on our theoretical framework, we expect LP-Robust Conformal to perform well in this domain.

---

> ### Author Response · Authors · 2025-08-04
> **Additional real-world regression experiment**
>
> To provide additional results on real-world distribution shift, particularly for regression tasks, we aimed to include the PovertyMap dataset in our experiments. However, accessing PovertyMap requires an approval process that may take several weeks. To avoid delaying the response, we selected the Shifts Weather Prediction dataset instead, which offers a high-quality regression task under naturally occurring covariate and domain shifts. We believe this dataset serves as a strong and realistic benchmark, and complements our existing evaluation by further demonstrating the robustness and generalizability of our method in real-world settings.
>
>
> **Summary of the dataset and regression task:**
>
> The Shifts Weather Prediction dataset defines a scalar regression task: predicting surface temperature from 123 heterogeneous meteorological covariates and 4 metadata attributes (latitude, longitude, time, and climate type). It contains 10 million entries collected from September 2018 to September 2019 across diverse climate zones, reflecting real-world distribution shifts. A canonical split defines in-domain data (Tropical, Dry, Mild Temperate) and out-of-domain test sets (Snow, Polar), enabling robust evaluation under temporal and geographic variation. In our experiments, we predict the temperature using a CatBoostRegressor ensemble with 10 members as the forward model, and use the absolute error between the predicted and true temperature as the conformity score. To assess robustness in this setting, we applied our proposed LP-robust conformal prediction method, alongside five benchmark approaches including standard split conformal prediction and distribution shift–aware baselines. We estimated the $(\varepsilon, \rho)$ parameters in our method using the algorithm described in Appendix A.
>
> | Method       | Coverage | Interval Length |
> |---------------|------------------|-------------------|
> | SC     | 0.771    | 4.754           |
> | Weight      | 0.783    | 4.892           |
> | $\chi^2$          | 0.855    | 5.902           |
> | RSCP          | 0.902    | 7.272           |
> | FG-CP  | 0.912    | 7.154           |
> | $\mathrm{LP}_\mathbf{\varepsilon}$     | **0.900**    | **6.824**           |
>
> **Explanation of results:**
>
> Classical conformal prediction (SC), covariate shift–aware weighting (Weight), and the $f$-divergence–based method ($\chi^2$) all exhibit undercoverage, as they fail to account for global distribution shifts. Among the methods achieving valid coverage—RSCP, fine-grained CP (FG-CP), and our LP-robust approach—ours yields the smallest prediction intervals. RSCP produces wider sets potentially due to smoothed scores, and FG-CP slightly overcovers due to challenges in estimating conditional divergences. In contrast, our method achieves both valid and sharp coverage, suggesting that the learned $(\varepsilon, \rho)$ pair effectively captures the shift.

---

> > ### Comment · Reviewer_Ro7g · 2025-08-04
> >
> > Thank you for your efforts to address my concerns! I maintain my positive score of 4.

---

### Official Review · Reviewer_odWU · 2025-07-03

**Clarity:** 4
**Significance:** 3
**Originality:** 3
**Rating:** 4
**Confidence:** 4

**Summary:**

This work considers the problem of conformal prediction under distribution shift.  Formally, they consider the problem where the training/calibration data is drawn i.i.d. from some distribution $\mathbb{P}$, and test data is drawn from another distribution $\mathbb{Q}$.  Formally they give a method such that if the $\varepsilon$-Levy-Pokhorov distance between the distribution $\mathbb{P}$ and the distribution $\mathbb{Q}$ is less than $\rho$, then the conformal predictor that is trained on $\mathbb{P}$ will exhibit coverage on $\mathbb{Q}$.  The $\varepsilon$-Levy-Pokhorov distance between $\mathbb{P}$ and $\mathbb{Q}$ is the minimum TV-distance between $\mathbb{Q}$ and any distribution $\widetilde{\mathbb{P}}$ that has $\infty$-Wasserstein distance $\le \varepsilon$ to $\mathbb{P}$.  Thier predictor works by taking a larger prediction set, which expands the quantiles of the score function covered by $\rho$ and expands in the original space by $\varepsilon$.

**Questions:**

1. This notion of robustness is defined for data that are drawn i.i.d., with the training drawn from distrbution $\mathbb{P}$ and the test data drawn from  a nearby distribution $\mathbb{Q}$.  Usually, in the standard conformal prediction model, the data is exchangeable, but not necessarily drawn i.i.d. from some unknown distribution.  Are there similar notions that can be defined for exchangeable data?

2. It is not immediately clear whether the exchangeability assumption of standard conformal prediction is stronger or weaker than the assumption that the Levy-Pokhorov distance between $\mathbb{P}$ and $\mathbb{Q}$ is bounded.  Is there a natural sense in which it is a stronger assumption?  That is, suppose you had $N$ exchangeable random variables $X_1, \dots, X_N$.  Is the Levy-Pokhorov distance between the distributions $\text{Unif}(X_1, X_{N/2})$ and $\text{Unif}(X_{N/2 + 1}, X_{N})$ small with high probability?  It would be helpful to provide an in depth comparison of these two models.

3. Could you interpret the dependence of the efficiency of your method on $\varepsilon$ and $\rho$?  What kinds of values of $\rho$ and $\varepsilon$ can you expect nontrivial results for?  As a starting point, to get a non-trivial result, one needs to assume that $\rho < \alpha$.  In what settings can we expect this to hold.  Suppose $\alpha = 0.1$, as it is in many applications of interest.  Then $\rho < 0.1$ means that you cannot consider any $\mathbb{Q}$ that has TV-distance more than $0.1$ from $\mathbb{P}$.  Is this a strong assumption?

4. Your method increases the size of the prediction set in a conservative way (hence the concerns above about efficiency).  Can you show that this is necessary to achieve this definition of robustness?  That is, can you show that, out of all possible conformal methods that achieve robustness in the Levy-Pokhorov sense, yours is the most efficient possible?  I think perhaps you can, if so it would be nice to include this as a formal statement.

- Line 152: Minor typo: should $W_\infty (\mathbb{P}, \mathbb{Q}) \le \rho$ have $\varepsilon$ instead of $\rho$?

**Ethical Concerns:**

["NO or VERY MINOR ethics concerns only"]

**Final Justification:**

I appreciate the authors thoughtful and thorough responses to my concerns.  My main concern remains, that while the observation that it is possible to raise the conformal score threshold in a way that corresponds to Levy-Pokhorov distance is very elegant and clean, it is very simple, and does not require a lot of technical work to prove.  Thus I maintain my score at 4: weak accept.

**Limitations:**

Yes

**Quality:**

2

**Strengths And Weaknesses:**

**Weaknesses** \
The technical result is extremely simple.  It is not surprising that increasing the coverage in the training stage makes coverage more robust in the testing stage.  It would nice to have some more theoretical investigation into this method: how much can it affect the efficiency of the method?  Is this the most efficient method possible that can achieve a distributionally robust guarantee?  (Please see questions for authors)

Also, it is not immediately clear to me how strong of an assumption bounding the Levy-Pokhorov distance is.  Is it possible to get non-trivial guarantees for moderate to large Levy-Pokhorov distances?  (See questions for more details)

**Strengths** \
The problem is very well-motivated.  A major drawback of standard conformal methods is that they assume exchangeability between the training examples and the test examples.  But this is a drawback in deploying this to real world settings, which almost always exhibit drift.  Previous works model specific ways in which the distribution can change, and design methods that fit to that.  However, this strategy requires knowledge of the drift, which is counter to the spirit of distribution-free guarantees.  The modeling of the problem in the framework of distributionally robust optimization is very elegant.

The numerical experiments are also compelling.  Since the method essentially just increases the coverage on the training set in a conservative way, the natural question is how it affects the efficiency of the method.  The numerical methods suggest that this method does not make the prediction sets too much bigger in natural settings.

---

> ### Author Rebuttal · Authors · 2025-07-30
>
> We are very grateful for your time and effort in reviewing our paper. Thank you for the positive assessment of our work. Below, we provide an answer to each of your questions (which also provide an answer to the weaknesses).
>
> ------
>
> **Question 1.** Thank you for this insightful question. The standard split conformal prediction framework relies on the assumption that the $n$ calibration scores and the $(n+1)$‑th test score are exchangeable, which in particular requires that they are drawn from the same distribution (though they may be correlated). Under LP distribution shifts, however, the distribution of the test score necessarily differs from that of the calibration scores, violating exchangeability and rendering the standard method inapplicable. In order to address this more general setting, we require the assumption that the calibration scores are i.i.d. from the calibration distribution. This assumption seems technically essential, as it enables us to control the discrepancy between the empirical calibration distribution and its true, unknown counterpart, a property that is used explicitly in the final step of the proof of Theorem 4.1.
>
> -----
>
> **Question 2.** Following the answer provided to the first question, the assumption of exchangeability between the $n$ calibration scores and the $(n+1)$‑th test score is strictly stronger than the LP distribution shift assumption that we impose (indeed, if the scores are exchangeable, then LP distance between the calibration and test distributions must be zero). In fact, since the test distribution and the calibration distribution are different, the standard proof reasoning based on rank statistics fails in the distribution shift case. We hope that this answers the reviewer's question, but are happy to provide more details in case our answer is not fully clear.
>
> ----
>
> **Question 3.** Thank you for this sharp question. As noted in Remark A.1 (Appendix A of the submitted version), we discussed the sensitivity of the coverage guarantee in Theorem 4.1 to the parameters $\varepsilon$ and $\rho$. However, we acknowledge that this remark does not directly address the statement in question 3 that “to get a non‑trivial result, one needs to assume that $\rho < \alpha$.” We emphasize that $\rho < \alpha$ is not an assumption required by any of our technical results. Rather, it excludes the degenerate scenario in which the confidence interval covers the entire real line. Indeed, if $\rho \geq \alpha$, one can construct a sequence of distributions whose $(1-\alpha)$‑quantiles diverge to infinity (see Remark 3.3 in the submitted version). Thus, the condition $\rho < \alpha$ is not a limitation of our analysis, but a reflection of the phenomenon itself.
>
> We also note that the condition $\rho < \alpha$ does not imply that the total variation distance is less than $\alpha$. Such an implication would hold only in the special case $\varepsilon = 0$, since any local perturbation also contributes to the total variation distance.
>
> -----
>
> **Question 4.** Thank you for the opportunity to clarify this point. Among all conformal methods that achieve robustness in the LP sense, we claim that ours attains the highest possible efficiency. This follows directly from the exact (i.e., approximation-free) results in Propositions 3.4 and 3.5, which characterize the worst‑case quantile and coverage, respectively. These exact characterizations imply that the inequality in the first display equation of the proof of Theorem 4.1 becomes an equality in the worst case. The only source of approximation in Theorem 4.1 arises in its final step, where we relate the $\beta$‑quantiles of the empirical calibration distribution to those of the true calibration distribution. This approximation appears technically unavoidable in the absence of full knowledge of the true calibration distribution.

---

> > ### Comment · Reviewer_odWU · 2025-08-06
> >
> > I appreciate the authors thoughtful and thorough responses to my concerns.  My main concern remains, that while the observation that it is possible to raise the conformal score threshold in a way that corresponds to Levy-Pokhorov distance is very elegant and clean, it is very simple, and does not require a lot of technical work to prove.  Thus I maintain my score at 4: weak accept.

---

### Decision · Program_Chairs · 2025-09-17

**Decision:**

Accept (poster)

**Comment:**

This submission proposes a robust extension of conformal prediction under distribution shifts using Lévy–Prokhorov (LP) ambiguity sets, which simultaneously capture local and global perturbations. The framework is natural, connects well to optimal transport, and yields closed-form characterizations of worst-case quantiles and coverage. While the main technical observation is relatively simple and may not push the theoretical frontier substantially, it is clean, well-motivated, and addresses a practical limitation of standard conformal methods. The experimental evaluation, though somewhat limited in scope, aligns with the theory and provides evidence that the approach maintains good coverage without excessive loss in efficiency. I lean toward acceptance.